# Terminal α1,2-fucosylation of glycosphingolipids by FUT1 is a key regulator in early cell-fate decisions

Saray Chen [1,5], Dana Hayoun-Neeman[1,5], Michal Nagar [1], Sapir Pinyan[1], Limor Hadad[1], Liat Yaacobov[1], Lilach Alon[1], Liraz Efrat Shachar [1], Tair Swissa[1], Olga Kryukov[1], Orly Gershoni-Yahalom [2,3], Benyamin Rosental [2,3], Smadar Cohen[1,2,4] & Rachel G Lichtenstein [1,2 ✉]

## Abstract

The embryonic cell surface is rich in glycosphingolipids (GSLs), which change during differentiation. The reasons for GSL subgroup variation during early embryogenesis remain elusive. By combining genomic approaches, flow cytometry, confocal imaging, and transcriptomic data analysis, we discovered that α1,2-fucosylated GSLs control the differentiation of human pluripotent cells (hPCs) into germ layer tissues. Overexpression of α1,2-fucosylated GSLs disrupts hPC differentiation into mesodermal lineage and reduces differentiation into cardiomyocytes. Conversely, reducing α1,2-fucosylated groups promotes hPC differentiation and mesoderm commitment in response to external signals. We find that bone morphogenetic protein 4 (BMP4), a mesodermal gene inducer, suppresses α1,2-fucosylated GSL expression. Overexpression of α1,2-fucosylated GSLs impairs SMAD activation despite BMP4 presence, suggesting α-fucosyl end groups as BMP pathway regulators. Additionally, the absence of α1,2-fucosylated GSLs in early/late mesoderm and primitive streak stages in mouse embryos aligns with the hPC results. Thus, α1,2-fucosylated GSLs may regulate early cell-fate decisions and embryo development by modulating cell signaling.

Keywords α1,2-Fucosylated Glycosphingolipids; Fucosyltransferase 1; Human Pluripotent Stem Cells; Mesoderm; BMP Signaling
Subject Categories Membranes & Trafficking; Signal Transduction; Stem Cells & Regenerative Medicine

## Introduction

During the early stages of embryonic development, cell-extrinsic and cell-intrinsic features guide pluripotent epiblast (EPI) differentiation, giving rise to the population of the three germ layers; mesoderm, endoderm, and ectoderm (Kojima et al, 2014).

Although the use of cell-extrinsic features in the characterization of the early stages of cell differentiation has been extensively researched, the investigating of intrinsic features' contribution to early fate choices is limited. Moreover, while signaling molecules in the embryonic cell microenvironment contribute to cell-to-cell heterogeneity during cell commitment, the cell-intrinsic properties that govern signaling molecules have not been fully studied (Cheng et al, 2022).

GSLs are a sub-family of lipids that are abundant in the membrane of eukaryotic cells and consist of a ceramide backbone glycosidically linked to oligosaccharides. The oligosaccharides are classified into globo-series (GalNAcβ3Galα4Gal), lacto-series (Galβ3GlcNAcβ3Gal; type 1), neolacto-series (Galβ4GlcNAcβ3Gal; type 2), and asialo- and ganglio-series (Galβ3GalNAcβ4Gal) and possess an assortment of sugar combinations (Handa and Hakomori, 2017; Russo et al, 2018a). The oligosaccharide compartment's structural variety is attributable to the widespread activity of many glycosyltransferases (GTs), which catalyze the addition of sugar residues to the lipid backbone or to another sugar acceptor along the secretion pathway (Handa and Hakomori, 2017). During embryogenesis, morphogens, which remodel gene expression programs associated with cell-fate decisions, may also reprogram GT genes and therefore play a role in determining GSL composition (Pecori et al, 2021). Consequently, different sub-types of GSLs produced in different stages of embryogenesis might play a pivotal role in specific stages, e.g., globo-series and lacto-series in the pre-implantation stage and ganglio-series in the early organogenesis stage (Ryu et al, 2017). In addition, both lipid and oligosaccharide compounds may interact with the membrane receptors and thus regulate receptor activity (Coskun et al, 2011; Liang, 2022) and signal transduction associated with a specific receptor. Therefore, GSL composition is an integral component of developmental programs, stem cell phenotyping, and stem cell differentiation (Capolupo et al, 2022).

Based on previous studies showing down-regulation of fucosyltransferase 1 (FUT1) during stem cell differentiation (Ojima et al, 2015), we hypothesize that α-fucosyl structures regulate cell signaling earlier in the time window between the pluripotency and differentiation states (24–72 h), when lineages of the three germ

[1]Avram and Stella Goren-Goldstein Department of Biotechnology Engineering, Faculty of Engineering Sciences, Ben-Gurion University of the Negev, Beer-Sheva 8410501, Israel. [2]Regenerative Medicine and Stem Cell (RMSC) Research Center, Ben-Gurion University of the Negev, Beer-Sheva 8410501, Israel. [3]The Shraga Segal Department of Microbiology, Immunology and Genetics, Ben-Gurion University of the Negev, Beer-Sheva 8410501, Israel. [4]The Ilse Katz Institute for Nanoscale Science and Technology, Ben-Gurion University of the Negev, Beer-Sheva 8410501, Israel. [5]These authors contributed equally: Saray Chen, Dana Hayoun-Neeman. ✉E-mail: ruha@bgu.ac.il

layers are formed after exposing hESCs to extrinsic triggers and where the major GSLs are globo-series and lacto-series (Russo et al, 2018a). FUT1 adds fucose residue in α1-2 linkage to terminal galactose of GSLs and glycoproteins. This study focuses on the α1,2-fucosyl structures of GSLs in pluripotency and early differentiation phases, as well as up to E7.5 embryos, where α1,2-fucosyl glycan structures are solely synthesized in GSLs (Kawamura et al, 2014, 2015; Ashwood et al, 2020).

# Results

## Changes in α1,2-fucosyl group expression occur during embryogenesis

Given that globo-series are mainly expressed during the pre-implantation and late gastrulation stages (Sato et al, 2007), we reasoned that Globo-H, as well as other globosides (Ojima et al, 2015), might be expressed by various groups of cells in the embryo as embryonic development progresses.

To determine whether this is the case, we isolated mouse embryos at several stages and examined the distribution of fucosyl groups and fucosyltransferase 1 gene and protein (FUT1, FT1, respectively). In a covalent fashion, FUT1 adds fucose residue to lacto-series (SSEA-5) and globo-series (α1-2 fucosyl-Gb5; Globo-H) (Fig. 1A). Immunofluorescence analysis for FT1 protein and Globo-H of wild-type C57BL/6 mouse embryos at pre- and peri-implantation stages (Bedzhov and Zernicka-Goetz, 2014) (Fig. 1B) showed Globo-H expression in the epiblasts (EPIs) of E5.25 embryos (Fig. 1C), confirming that α1-2 fucosyl GSLs predominate in pluripotent cells. Mouse embryos in the post-implantation stage (E7.5) expressed both FUT1 and Globo-H in highly restricted regions within cell clusters of the definitive endoderm (DE) in the anterior and posterior primitive streak (PS) according to Brachyury (BRY), MIXL1, and CDX2 expression (Fig. 1D). In the organogenesis stage (E8.75), staining for α-fucose residues, using Ulex Europaeus Agglutinin I (UEA I), exposed scattering of α1-2 fucosyl glyco-conjugates in the gut tube (Gt) and hindbrain (Hb) in somite position based on BRY (Fig. 1A,E). We were unable to detect the presence of α-fucosyl structures in the heart tube (H) or in the cells of several other tissues composing the developing organs (Fig. 1E), confirming that during the commitment to a specific fate, most of the cells stop synthesizing α1,2-fucosyl GSL, apparently to regulate signaling events (Guri et al, 2017; Park et al, 2017).

Given all the above, we therefore wondered whether pluripotent EPIs destined to become cardiac tissue terminate α1,2-fucosyl glycoconjugate synthesis upon heart development. To address this, we surveyed a dataset of RNA-seq analyses observed during mouse cardiogenesis (Li et al, 2014) (GEO series accession number GDS5003), to identify GT genes co-expressed with cardiac markers. We found that the expression of genes associated with heart development (Estarás et al, 2017), as well as gene expression of GTs involved in the synthesis of ganglio-series (Russo et al, 2018a), were both upregulated over time, while the expression of many genes associated with globo-series and lacto-series synthesis was downregulated in heart development. Specifically, FUT1 and SEC1 gene expression was significantly downregulated by cells at PS stage (E7.5) relative to the expression

in the pluripotency state (Fig. 1F). Transcripts of SEC1, a GT defined only in mice and catalyzes α1,2-fucosyl reaction to glycoproteins (Domino et al, 2001), were marginally generated during cardiogenesis. However, FUT2, the FUT1 paralog (Chang et al, 2008), was present but displayed a different genetic expression pattern than FUT1 in heart development (Fig. 1F), indicating that FUT1 and FUT2 might catalyze fucosyl reaction to distinct structures or be active in different lineages of various sections of the developing heart.

We then searched for a model that would show the spatiotemporal expression of the FUT1 gene in embryos at an early stage of development, before emergence of the PS. We found GEO series accession number GSE120963 from the publicly available datasets of Peng et al (Peng et al, 2019) provided a suitable dynamic transcriptomic model of mouse embryos in the pre- and late-gastrulation stages. Based on these data, we performed single-cell RNA-seq and GEO-seq analyses for FUT1, FUT2, and SEC1 gene expression. Because the identity of naive pluripotent EPIs in the inner cell mass (ICM) of the blastocyst is established during the first 4 d of development (Guo et al, 2021; Bergmann et al, 2022), we analyzed the co-expression of FUT1 with lineage-specific markers as well as pluripotent genes.

Two-dimensional (2D) corn plots of the dynamic expression of FUT1 transcripts in defined locations in the mouse embryos (E5.5–E7.5) exhibit a small number of early and late EPIs and endoderm cells expressing different levels of FUT1 transcripts (Fig. 1G). We observed over 95% co-expression of FUT1 with OCT3/4, NANOG, and SOX2 in EPIs (E6.0) (Fig. EV1A). As seen (Fig. EV1B; Appendix Fig. S1A), clusters of anterior–posterior EPIs (E6.0 and E7.0) expressing FUT1 and ectoderm-specific genes are finally fated to form ectodermal layer cells (E7.5). This is consistent with the embryo staining for α1-2 fucose residues (E7.5), where FUT1 is co-expressed with OTX2 and DBX1 (Appendix Fig. S1A), two neuroectoderm (NE) markers (Gouti et al, 2014; Metzis et al, 2018). In addition, the FUT1 gene is partially expressed in the distal endoderm domain (E7.0), which is populated exclusively by E1, an endoderm population expressing the transcription factors (TFs) SOX7, SOX17, and LEFTY1. At E7.5, the FUT1 gene is partially expressed in clusters of allocated, distinct endodermal lineages E1, E2, and E3 (Fig. 3 in (Peng et al, 2019) and Fig. EV1C). Notably, FUT1 seldom overlapped with canonical mesoderm markers (E7.5) (Fig. EV1D). Two clusters of PS cells in the posterior distal region (E7.0) express FUT1, but these clusters are not aligned with the PS (E6.5–E7.5) or related to the mesoderm (E7.5). However, one cluster that consists of mesectoderm (Bellefroid et al, 1998) and mesendoderm (Legier et al, 2023) co-expressed FUT1 at lower levels with IRX3, HOPX, and CDX2 transcripts. This model demonstrates the dynamic scattering of FUT1 during the peri- and post-implantation periods of mouse development, implying that FUT1 is an important gene in the gene array that defines EPI identity, despite not being involved in the formation of the mesoderm. In other words, the choice to downregulate the FUT1 gene is part of embryonic patterning at gastrulation. Moreover, this model's consistency with embryo staining highlights that FUT1 activity or the synthesis of α1,2-fucosyl structures is not essential in establishing cardiac identity (E8.75) (Fig. 1E).

We then analyzed the expression of FUT2 and SEC1 within the embryos and found that roughly 10% of FUT2 and SEC1 mRNA

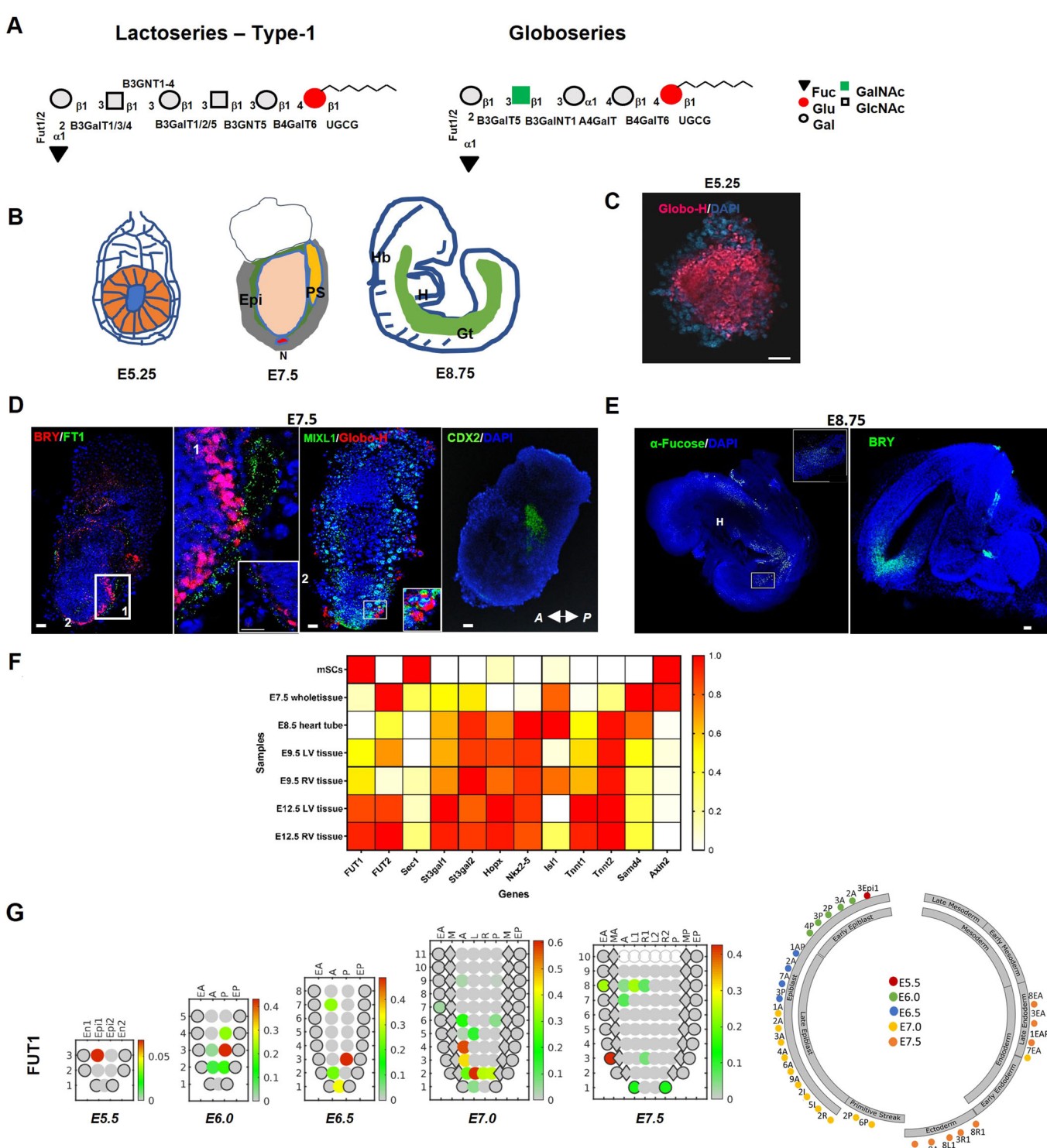

**A**

**Lactoseries – Type-1**

**Globoseries**

**B** E5.25    E7.5    E8.75

**C** E5.25 — Globo-H/DAPI

**D** E7.5 — BRY/FT1 | MIXL1/Globo-H | CDX2/DAPI

**E** E8.75 — α-Fucose/DAPI | BRY

**F** (heatmap)

Samples: mSCs, E7.5 wholetissue, E8.5 heart tube, E9.5 LV tissue, E9.5 RV tissue, E12.5 LV tissue, E12.5 RV tissue

Genes: FUT1, FUT2, Sec1, St3gal1, St3gal2, Hopx, Nkc2-5, Isl1, Tnnt1, Tnnt2, Samd4, Axin2

**G** FUT1 — E5.5, E6.0, E6.5, E7.0, E7.5

E5.5, E6.0, E6.5, E7.0, E7.5

overlapped with FUT1 (E5.5–E7.5). FUT2 transcripts were detected in cell clusters of the posterior mesoderm (E6.5), and later in the posterior and anterior mesoderm, but not in the distal region of the mesoderm (E7.0–E7.5) (Appendix Fig. S1B). SEC1 transcripts were abundant (E5.5–E6.0) and were detected in restricted regions as FUT2, and in one cluster in the distal part of the embryo (Appendix Fig. S1C).

With responsiveness to the finding of Peng et al (Peng et al, 2019) their expression during early cardiogenesis, the two fucosyltransferase homologs, FUT1 and FUT2, apparently catalyze a reaction to produce α-fucosyl structures with different GSL compositions, however, SEC1 is probably needed in specific regions. Nevertheless, more evidence is needed to test this hypothesis.

**Figure 1. FUT1 expression is reduced during early gastrulation and cardiogenesis.**

(A) Symbolic representation of lacto-series type 1 and globo-series showing synthesis consequences with corresponding GTs. The GTs are UGCGT1, B3GalNT1, B3GNT5, B3GalT2, FUT1/FUT2, ST3Gal1/2, and FUT3 (reagents and tools table and Appendix Fig. S2), and the sugar residues are Fuc, fucose; Glu, glucose; Gal, galactose; GalNAc, N-acetylgalactosamine; and GlcNAc, N-acetylglucosamine. (B) Representative whole embryos 5.25, 7.5, and 8.75 days after breeding (E5.25, E7.5, and E8.75, respectively). Abbreviations represent EPI, epiblast; N, notochord; PS, primitive streak; H, heart; Hb, hindbrain; Gt, gut tube. (C) Representative immunofluorescence staining of whole embryo for Globo-H (red) on E5.25. (D) Representative immunofluorescence staining of whole embryo for BRY (red), MIXL1 (green), Globo-H (red), CDX2 (green), mouse FT1 (green) on E7.5. (E) Representative immunofluorescence staining of whole embryo for BRY (green) and FT1 (green) on E8.75. Nuclei are stained with DAPI (blue). (F) Heatmap showing a subset of RNA-seq-based expression profiles of GT genes, cardiac markers, and Wnt- and BMP-responsive genes derived from dataset of mouse stem cells (mSCs), E7.5 whole-tissue embryos, E8.5 heart tube, and E9.5 left ventricle (LV) and right ventricle (RV). Dark orange indicates high expression; white indicates low expression of fold change on logarithmic scale ($n = 3$ mice). (G) Left: 2D corn plots showing the spatiotemporal expression of FUT1 in E5.5–E7.5 embryos. Right: Circle diagrams demonstrating FUT1 expression according to tissue classification. En1 and En2, divided endoderm; Epi1 and Epi2, divided epiblast; EA, anterior endoderm; EP, posterior endoderm; A, anterior; P, posterior; M, whole mesoderm; L, left lateral; R, right lateral; L1, anterior left lateral; R1, anterior right lateral; L2, posterior left lateral; R2, posterior right lateral; MA, anterior mesoderm; MP, posterior mesoderm. Data information: In (C), scale bar represents 20 µm. In (D, E), scale bars represent 100 µm.

## Differentiation of hESC lines into cardiac cells involves a dynamic expression of GTs

We then examined whether α1,2-fucosylated glycoconjugate patterning during cardiogenesis, was a feature of human cardiac development.

To address this, we used an established in vitro model of human cardiomyocytes (CMs) based on early signaling and repression of WNT/β-catenin in hESC cultures (Fig. 2A) (Hayoun-Neeman et al, 2019; Lian et al, 2013). We can see that differentiated cells (day 2), cardiac progenitors (day 5–6), and immature CMs (day 7) (Fig. 2A,B) downregulated FUT1 and expressed low FT1 protein (Fig. 2C). FUT1 transcripts were re-expressed at a significant level in immature CMs from day 13 to differentiation to day 21 (Fig. 2C), indicating a rise in FUT1 mRNA towards CM maturation (Fig. 2A). These results are in line with FUT1 expression in mouse cardiogenesis (Fig. 1F). Gene expression of FUT2, however, was upregulated under WNT inhibition (48 h after CHIR was withdrawn (Fig. 2A,C)) and continued to rise during differentiation, while FUT1 transcripts decreased, similar to the pattern found between FUT1 and FUT2 gene expression in early mouse cardiogenesis (Fig. 1F).

We then tested whether the differentiated cells regulate other GTs involved in GSL synthesis during cell commitment to CMs. GEO series accession number GSE48257 from the publicly available datasets (Gu et al, 2014) highlighted a cluster of genes implicated in GSL synthetic pathways. Many of the genes within this cluster ($p < 10^{-4}$) are associated with globo-series and lacto-series GSL synthesis, including B3GalNT1, B3GNT5, B3GalT2, FUT1, FUT2, and FUT3 (Russo et al, 2018a), and their expression was downregulated 14 d after hESCs' commitment to CMs. Other GT genes involved in the ganglio-series synthesis, including ST3Gal1 and ST3Gal5, were upregulated in day 14 CMs relative to stem cells (Appendix Fig. S2). These results were confirmed by qPCR analysis of several GT genes measured during 21 d of hESCs' differentiation into CMs. We observed upregulation of gene expression of some GTs (B3GNT, FUT2, FUT3, ST3Gal1, and ST3Gal5), as well as a pattern of circadian-like oscillation expression of the other GTs (UGCG, B3GalNT1, and B3GalT2) (Fig. 2C), confirming that the developmental process is accompanied by GSL metabolic reprogramming (Russo et al, 2018b).

Notably, the downregulation of the FUT1 gene on days 3–5 is apparently linked with a 50-fold upregulation of HOPX transcripts on day 3 under CHIR withdrawal and a 3-fold upregulation of

SMAD4, one of the BMP transduction proteins, concurrent with WNT repression, on day 5 (Fig. 2D), suggesting that FUT1 downregulation and α-fucosyl group reduction are essential to myogenesis (Jain et al, 2015).

## Downregulation of FUT1 is one of the hallmarks of early differentiation

Pluripotent stem cells (PSCs) express GSLs of globo- and lacto-series, particularly; PSCs express α-fucosylated structures, which mainly decrease upon differentiation into embryonic bodies (EBs) and neuronal progenitors (Liang et al, 2011).

We therefore investigated whether differentiation protocols incorporating combinations of extrinsic triggers that drive hESCs into distinct progenitor fates would change the expression of FUT1, FT1 and the α-fucosyl chains it synthesizes during cell differentiation. We initially generated EBs by performing hESC differentiation in medium supplemented with 10% fetal bovine serun (FBS) for 5 days (Fig. 3A). EBs in 5 d cultures downregulated embryonic markers (Appendix Fig. S3A,B) and expressed genes of the tri-lineages (Appendix Fig. S3B), including α-fetoprotein (α-FP, an endoderm marker, 25% ± 0.95%) and the mesoderm marker, HAND1, (29.9% ± 1.4%), but no detectable posterior determinant, CDX2 (Fig. 3C). As expected, the number of FT1$^+$OCT4$^+$ cells declined steadily (91% ± 1.0% on day 0), while the number of FT1$^-$OCT4$^-$ cells increased (71.5% ± 2.6%) (Fig. 3B). Consistent with the FT1 observation in the EBs, we saw a 5-fold reduction of α-linked fucosyl residues in the EBs compared to the hESCs (Fig. 3D), implying that high expression of α-fucosyl residues may be necessary for keeping cells in the pluripotency state.

The use of two established protocols for differentiation of hESCs into DE (Loh et al, 2014; Hinton et al, 2010) (Fig. 3E) resulted in the downregulation of stem cell markers, OCT3/4 (28.8% ± 3.1%), NANOG (69.7% ± 1.1%) and SOX2 gene expression, as well as in the upregulation of canonical differentiation markers, including the endodermal-specific markers, FOXA2, SOX17, HHEX, and α-FP (52.8% ± 2.6%) (Fig. EV2A,B; Appendix Fig. S3C). Low expression of FT1 protein (60.2% ± 0.3%) compared to the pluripotent cells (89.6% ± 0.7%) and of FUT1 transcripts (10% relative to the pluripotent cells) was documented (Fig. 3F), although there was an expected similarity between differentiated and undifferentiated cells in the expression profile of FUT1 (Liang et al, 2011; Ojima et al, 2015). These results suggest that specific extrinsic triggers drive FUT1 downregulation within the cells during differentiation.

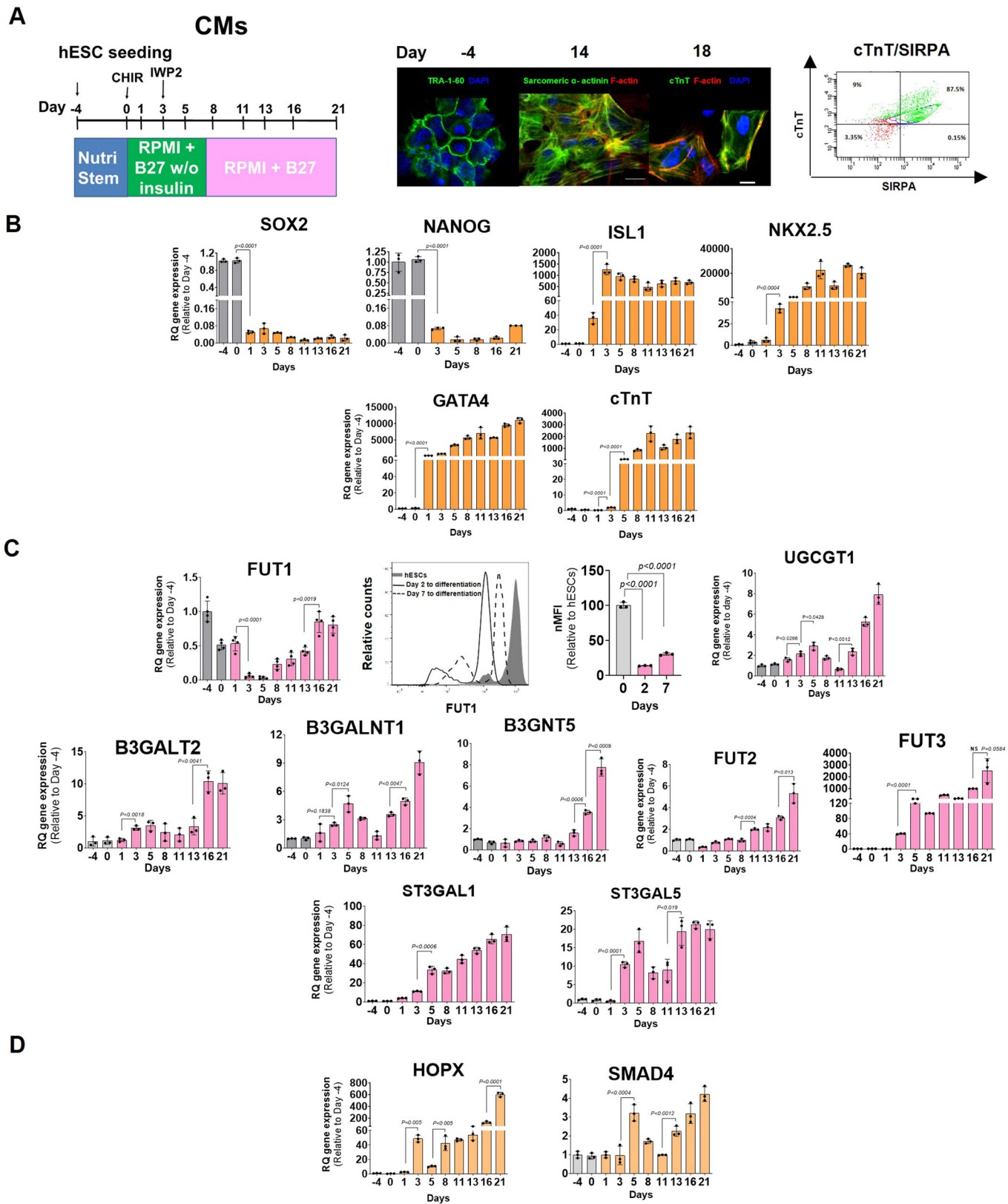

◄ **Figure 2. Profiles of GT gene expression are time dependent during cardiogenesis.**

(A) Left: Schematic showing differentiation of hESC into CMs via the temporal modulation of canonical Wnt/β-catenin from day 0 to day 21 in RPMI supplemented with modified B27 or complete B27. Center: Representative images of TRA-1-60 on day -4 and cardiac-specific markers (sarcomeric α-actinin and cTNnT) on day 14 and day 18 to differentiation. Right: Quantification of the percentage of CMs, showing the expression of proteins SIRPA and cTnT (green). Nuclei are stained with DAPI (blue) and F-actin with phalloidin (red). (B) Quantification of pluripotency genes (SOX2 and NANOG) and cardiac-specific markers (ISL1, NKX2.5, GATA4, and cTnT) during 21 d of hESC differentiation into CMs ($n = 5$ technical replicates). (C) Top left: Quantification of FUT1 during 21 d of hESC differentiation into CMs. Top right: Quantification of FUT1 expression on day 2 and day 7 of hESC differentiation into CMs ($n = 3$ technical replicates). Bottom: Quantification of gene encoding to GTs that synthesize lacto-series and globo-series (UGCGT1, B3GalNT1, B3GNT5, B3GalT2, FUT2, ST3Gal1, and FUT3) during 21 d of hESC differentiation into CMs (reagents and tools table and Appendix Fig. S2). (D) Quantification of HOPX and SMAD4 genes, which are involved in myogenesis ($n = 3$ technical replicates). qPCR data were normalized to values observed for hESCs on day $-4$. Data information: In (A), scale bars represent 20 μm. In (B–D), data are presented as means ± SEM. Two-tailed Student's t-tests $*p < 0.05$, $**p < 0.01$, $***p < 0.001$, $****p < 0.0001$.

Applying a single-step method for neural induction (Lee et al, 2010; Qu et al, 2014) efficiently induced hESCs into day 3 primitive neural progenitor cells (pNPCs) and day 6 NPCs (Fig. EV2C) and resulted in the upregulation of neural genes, PAX6, OTX2, and SOX1, and downregulation of the characteristic pluripotent marker, OCT3/4 (Fig. EV2C). In line with previous reports (Liang et al, 2011; Ojima et al, 2015; Russo et al, 2018b), FUT1 transcript, protein and α-fucose residues were reduced in neural progenitors relative to stem cells (Fig. 3G), highlighting the similarity in the gene expression signature and activity of FUT1 in mesoderm, endoderm, and ectoderm cells.

Next, we examined whether hiPSCs resemble hESCs by expressing a high copy number of FUT1 mRNA, although iPSC and ESC lines showed transcriptional pattern variability (Choi et al, 2015). Various GT genes quantified using qPCR in hESCs (H9.1) and hiPSCs (BJ, EMF, OME lines), underscored, to a certain extent, line-to-line variability. The hiPSCs expressed the GSL SSEA-4 and genes associated with stem cells in a slightly different manner than that observed in hESC lines (Appendix Fig. S3D). Somatic cell reprogramming resulted in a 400-fold downregulation of vimentin gene expression (Appendix Fig. S3D). The embryonic cells, reprogrammed cells, and somatic cells expressed similar and high FUT1 mRNA and protein, 98.0% ± 1.7% of OCT3/4$^+$ hESCs, 96.5% ± 2.7% of OCT3/4$^+$ hiPSCs, and 97.6% ± 1.0% of the OCT3/4$^-$ somatic cells (Appendix Fig. S3D; Fig. EV2D). The activity of FT1 corresponding to its gene expression was confirmed by the high level of α1,2-fucose residues found in FT1$^+$ hESCs (83.4% ± 1.0%) and FT1$^+$ hiPSCs (70.5% ± 1.4%) (Fig. EV2E). However, the low expression of α1,2-fucose residues in FT1$^+$ somatic cells (2.69% ± 0.04%) (Fig. EV2E) suggests that low α1,2-fucosyl structures catalyzed by FT1 might be necessary for hiPSC differentiation. The mean fluorescence intensity (MFI) of FT1 and α1-2 fucose residues, which was significantly higher in the stem cells than in the somatic cells, might indicate that low expression of FUT1 and low activity of FT1 occur during the process of PSC differentiation (Fig. EV2D,E). In addition, a 2-fold downregulation in the FUT2 gene expression of the reprogrammed cells relative to the embryonic cells (Appendix Fig. S3D), implying that FT1 but not its paralog is presumably a crucial enzyme required to catalyze α1,2-fucosyl structures in pluripotent cells.

We then tested whether hiPSC differentiation can also lead to FUT1 downregulation as differentiated hESCs. We used the hiPSCs to generate EBs for 5 d and found similar expressions of α-FP (45.6% ± 2.5%), HAND1 (23.4% ± 1.6%), and undetectable CDX2 protein (Fig. 3H) corresponding to hESC-derived EBs.

Consistent with the EBs from hESCs, most FT1$^+$OCT4$^+$ hiPSCs decreased from day 0 (80.7% ± 6.7%), while FT1$^-$OCT4$^-$ EBs increased on day 5 (69.9% ± 0.1%) (Fig. 3I). The 2-fold lower expression of α1-2 fucose residues in hiPSC-derived EBs compared to hiPSCs (in terms of the MFI) (Fig. 3J) showed a similar pattern of FUT1 expression and activity in hiPSCs and hESCs. These observations suggest that low FT1 expression and activity, possibly driven by extrinsic triggers, are essential for PSC commitment to distinct progenitor fates in earlier stages of differentiation.

## Pluripotency is associated with high levels of α 1,2-fucosyl GSLs

Core transcription factors of a complex transcriptional regulatory network like OCT3/4 and NANOG are essential for maintaining ESC pluripotency. Knockdown of these master genes below a threshold level leads to a disruption of the pluripotency state and promotes ESC differentiation (Niwa et al, 2000; Pan and Thomson, 2007; Heurtier et al, 2019; Xiong et al, 2022).

If the hypothesis that α1,2-fucosyl globo-series and lacto-series are hallmarks of PSCs, then the expectation would be that stem cells in which the FUT1 gene has been silenced would change cell morphology, downregulate genes characteristic of pluripotent cells, and upregulate lineage-specific markers. To test this, we knocked down FUT1 temporarily in three lines of hESCs (WA09, WIBR1, and WIBR2 (Lengner et al, 2010)) using several sequences of small interfering RNA (siRNA) designed to suppress FUT1 activity and two scrambled RNA interference sequences, which served as the control (non-targeting hESCs) and cultured the cells in NutriStem medium. Before knockdown, the three lines expressed similar α-fucosyl residues (Fig. EV3A), and the silencing resulted in FUT1 downregulation and low expression of α-fucose residues (Figs. 4A and EV3A). As expected for synthesis of α-fucosyl structures (Russo et al, 2018a), the fucose residues, the antigens, SSEA-5 (fucosyl-Lc4) and Globo-H (fucosyl-Gb5), decreased (Figs. 4B and EV3B,C), and SSEA-3 (Gb5) increased (Fig. 4B). Gene encoding for the glyco-synthesizing enzymes, B3GNT5 and B3GalNT1, which catalyze reactions for producing lacto-series and globo-series, respectively (Hakomori, 2007; Russo et al, 2018a) (Fig. 1A), were unaffected by the FUT1 silencing (Fig. 4C). However, the FUT2 gene, which was upregulated on days 2.25–3.25 (Fig. EV3D), did not catalyze the synthesis of α1,2-fucosyl antigens instead of the silenced FUT1 (Fig. EV3C), confirming that in hESCs, the α1,2-fucosyl embryonic antigens, like SSEA-5 and Globo-H, are products of

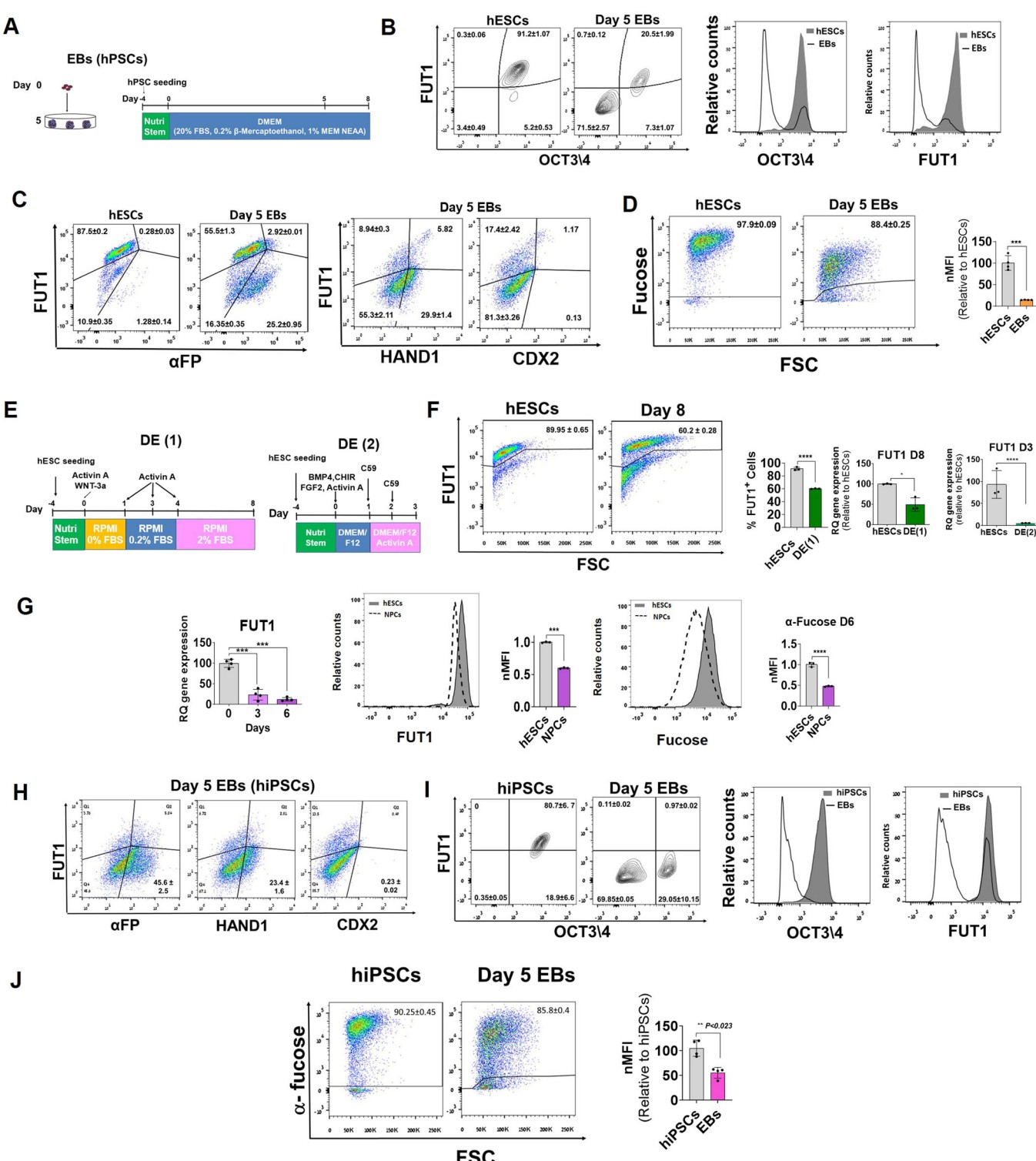

the FT1 enzyme. FUT1-silenced hESCs maintained morphology typical to undifferentiated hESCs. Cells continued to proliferate and exhibited similar cell cycle-associated markers, MKI67 and Cyclin D1, as untreated and non-targeting hESCs (Fig. 4A,D). However, the silenced cells showed a coordinated decrease in pluripotent gene expression, OCT3/4 and NANOG of less than

50% for a short period of 30–54 h, except the SOX2 gene (Fig. 4E), suggesting that the expression of α1,2-fucosyl groups by pluripotent cells does not imply pluripotency (Andrews and Gokhale, 2024).

In contrast, downregulation of FUT1 in hESCs led to a transient upregulation of lateral mesoderm (LM; cardiac) specific genes,

◄ **Figure 3.  Downregulation of FUT1 is one of the hallmarks of early differentiation.**

(A) Schematic showing hESC differentiation towards EBs over 5 d in a dish. (B) Representative histograms showing percentages and relative counts of hESCs and day 5 EBs express FT1 OCT3/4. (C) Quantification of the percentage of hESCs and day 5 EBs showing the expression of FT1 with DE- (α-fetoprotein (α-FP)) and mesoderm- (HAND1 and CDX2) specific markers. (D) Right at left: Quantification of the percentage of hESCs and day 5 EBs showing the expression of α-fucose relative to hESCs. Right at right: MFI of hESCs and day 5 EBs ($n = 3$ biological replicates). (E) Schematic representation of hESC differentiation protocols into DE; the first, DE (1), is via Activin-A and Wnt3a, followed by FBS, and the second, DE (2), is via bFGF, CHIR99021, and Activin-A, followed by the Wnt inhibitor C59. (F) Left: Quantification of the percentage of hESCs and hESCs-derived DE obtained by protocol 1 showing the expression of FUT1 in histograms relative to hESCs. Center: Quantification of FUT1 positive cells in hESCs-derived DE obtained by protocol 1 relative to hESCs. Right at left: Quantification of FUT1 transcripts in hESCs and hESCs-derived DE on day 8, obtained by protocol 1. Right at right: Quantification of FUT1 transcripts in hESCs and hESCs-derived DE on day 3, obtained by protocol 2. $n = 3$ technical replicates. (G) Left: qRT-PCR analyses of FUT1 in primitive NPCs; NPCs demonstrate a reduction of FUT1 with differentiation. Center at left: Representative histograms and MFI showing FT1 in primitive NPCs and hESCs. Right: Representative histograms and MFI showing α-fucose groups in primitive NPCs and hESCs. ($n = 3$). (H) Quantification of the percentage of hiPSCs and hiPSCs-derived EBs showing the expression of FT1 with DE (α-FP) and mesoderm- (HAND1 and CDX2) specific markers. (I) Representative histogram showing percentage and relative counts of hiPSCs and day 5 hiPSCs-derived EBs expressing OCT3/4 and FT1. (J) Representative histogram showing the percentage and MFI of hiPSCs and day 5 hiPSCs-derived EBs positive for α-fucose groups ($n = 4$). qRT-PCR data are normalized to the values of day-0 hiPSCs. Data information: In (B, C, D, E, F, G, H, I, J), data are presented as means ± SEM. Two-tailed Student's t-tests $*p < 0.05$, $**p < 0.01$, $***p < 0.001$, $****p < 0.0001$.

HAND1, HOPX, and SMAD4, except the NKX2.5 gene, in both the mRNA and protein levels for 30–102 h (Figs. 4F and EV3E). However, the elevation was several orders of magnitude lower than the mRNA copy number observed after hESC differentiation into LM (Loh et al, 2016) (Appendix Fig. S4A). These cardiac markers are early commitment genes of generated first heart field (FHF) progenitors, and most of them are strongly affected by BMP signaling (Klaus et al, 2007). Other LM genes, ISL1 and AXIN1, which are second heart field (SHS) progenitor markers and WNT-induced genes (Klaus et al, 2007; Jain et al, 2015) and the endoderm (FOXA2) and ectoderm (PAX6) markers were unaffected by FUT1 downregulation (Fig. 4F,G). In addition, FUT1-silenced hESCs did not express paraxial mesoderm (PM) markers (MSGN1 and CDX2) at comparable levels to day 2 hESC-derived PM (Loh et al, 2016) (Appendix Fig. S4G; Fig. 4G), suggesting that reduced α1,2-fucosyl residues may facilitate BMP signals involved in the formation of certain BMP-driven progenitors such as early heart progenitors (Zhang et al, 2023).

Furthermore, we note that hESCs with reduced FUT1 expression after RNA interference treatment did not lose pluripotency in the long term (Appendix Fig. S4C) or the ability to differentiate. When differentiating FUT1-silenced hESCs, untreated hESCs (in conditioned medium, OM), and non-targeting hESCs into CMs for 6 d, an equal number of OCT3/4$^+$ cells (5.97% ± 1.32%), NKX2.5$^+$ cells (87.5% ± 3.06%) and cTnT$^+$ cells (50.67% ± 3.72%) was seen in all lines corresponding to a cardiac progenitor identity (Appendix Fig. S4E). Notably, when FUT1-silenced hESCs were differentiated for 2 d to LM, FUT1 expression decreased to a low expression level. However, the number of mRNA copies in the silenced cells was 3-fold higher than observed in day 2 differentiated hESCs (Appendix Fig. S4D), indicating that FT1 levels decreased significantly during differentiation.

These results suggest that PCs exhibit significant levels of α1-2 fucosyl GSLs and these levels decreased significantly during differentiation, however, these GSLs do not indicate pluripotency. The findings are consistent with triple knockout homozygous FUT1/FUT2/SEC1 mice's capacity to produce viable and fertile mice (Chen et al, 2023). It remains to be seen whether the high expression level of α1,2-fucosyl GSLs prevents signaling pathways crucial for stem cell differentiation.

## Constitutive α1,2-fucosyl GSL expression impairs hESC commitment

As a result, we wondered if a constant high expression of α1,2-fucosyl residues could hinder hESC commitment to mesoderm cells.

To test this, we established three stable hESC lines expressing FUT1 and three stable mock-transfected hESC lines (control ECs). The transduction led to a remarkable induction of FUT1, FT1, α1,2-fucosyl residues, Globo-H and SSEA-5 in the FUT1$^+$ECs (Figs. 5A and EV4A). During culture under pluripotent conditions, a coordination of sequential hallmark events occurred: pluripotent stem-cell-associated markers as well as lineage-specific markers were maintained and cells retained the embryonic morphology along passages (Figs. 5B, EV4B, and EV4C). This led us to question if downregulation of FUT1 in hESCs could occur earlier, on days 1–2, to encourage hESC differentiation towards mesoderm progenitors (Loh et al, 2014, 2016) (Appendix Fig. S4A,B). Analysis of FUT1, FUT2, ST3Gal1, and ST3Gal2 genes using the datasets of Loh et al (GEO series accession numbers GSE85066 and GSE52657 (Loh et al, 2014; Loh et al, 2016)) revealed marked downregulation of FUT1 and FUT2 on day 1 anterior primitive streak (APS), day 1 mid-PS, and day 2 LM, PM, DLL$^-$PM, and DLL$^+$PM (Fig. EV4D), suggesting that the protocol of Loh et al suitable for testing LM-specific genes from day 1 to cell differentiation. Therefore, we examined how overexpression of the FUT1 gene affects LM-associated markers over three days of differentiation (Fig. EV4E). The expression of MESP1, FOXF1, HAND1, NKX2.5, IRX3, and HOPX genes was compromised on days 1–3 in FUT1$^+$ECs than their counterparts, as measured by qPCR and Fluorescence-activated cell sorting (FACS) (Fig. 5C–E). However, BRY and ISL1 genes were unaffected by FUT1 overexpression. As seen previously (Figs. 2D, and 4F), the particular LM genes that are compromised by the high α1,2-fucosyl glycoconjugate levels are downstream markers of BMP signaling (Astorga and Carlsson, 2007; Gunne-Braden et al, 2020; Jain et al, 2015). These observations suggest a regulatory circuit for α1,2-fucosyl glyco-conjugates and BMP downstream transcriptional responses.

In addition to LM markers, the expression of DE and NE genes and proteins (FOXA2, SOX17, PAX6) was slightly compromised in cells overexpressing FUT1 (Figs. 5F,G and EV4F,G), but the levels of other tested genes (SOX1, OTX2) remained unchanged in the

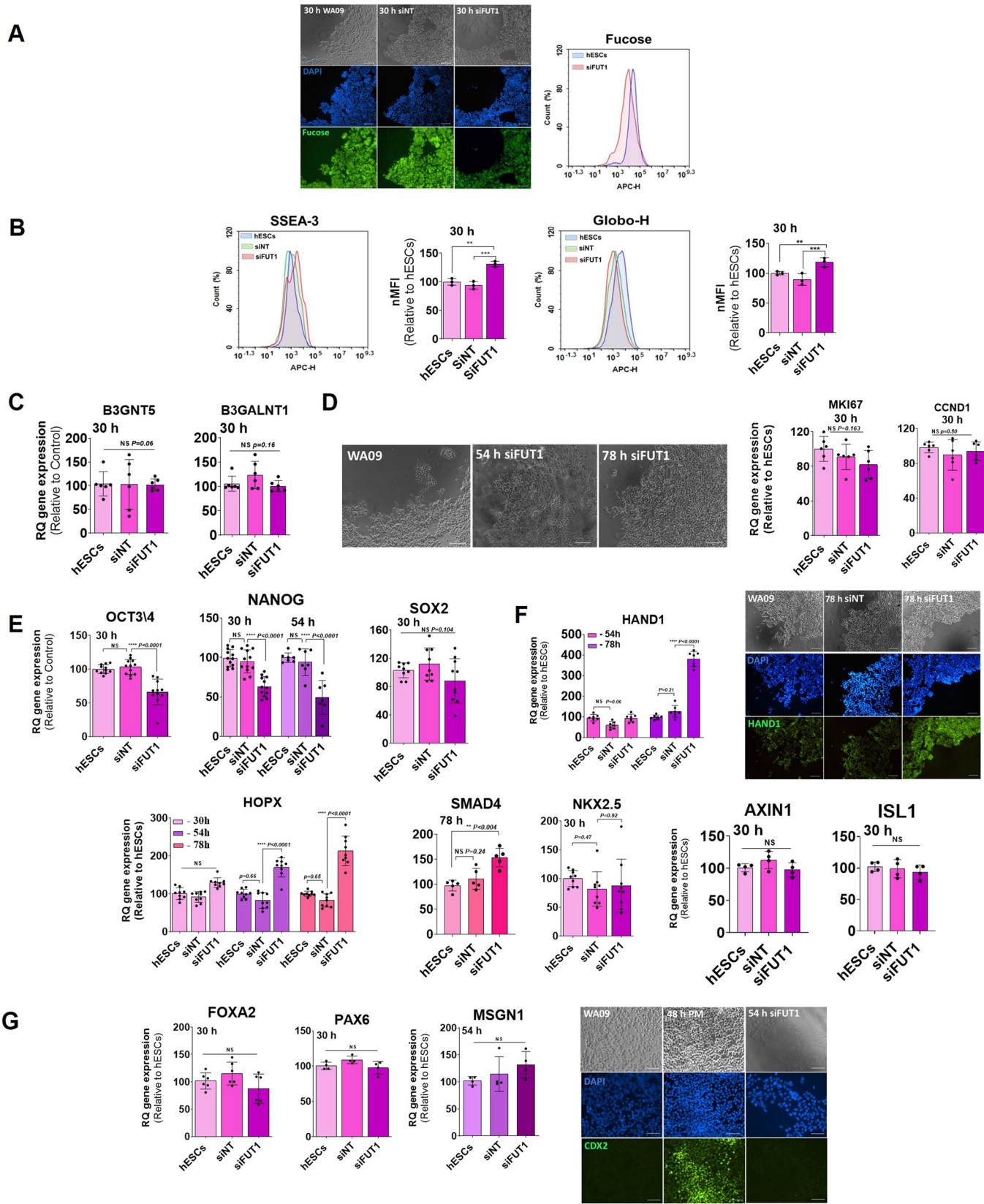

**Figure 4. BMP-induced transcription factors, characteristic of the first heart field, are affected by α1,2 fucosyl structures.**

(A) Left in top: Representative bright field images showing wild-type (WT) hESCs, siRNA non-targeting (siNT) and FUT1-silenced (siFUT1) hESCs. Left in center: Representative images showing nuclei of WT, siNT and siFUT1 hESCs stained with DAPI (blue). Left in bottom: Representative images showing α-fucose after staining WT, siNT and siFUT1 hESCs with UEA-I (green). Right: Representative histograms showing counts of siFUT1 hESCs expressing α-fucose relative to WT hESCs 30 h after mRNA silencing. (B) Left: Representative histograms showing counts of siNT and siFUT1 hESCs expressing SSEA-3 and MFI relative to WT hESCs 30 h after silencing. Right: Representative histograms showing counts of Globo-H expressed by siNT and siFUT1 hESCs and MFI relative to WT hESCs 30 h after silencing. WA09 hESCs were silenced with a mixture of 3 siRNA for FUT1 and 1 NT siRNA. $n = 3$ technical replicates. $n = 2$ technical replicates for $n = 3$ lines. (C) Quantification of B3GNT5 and B3GalNT1 transcripts from WT, siNT and siFUT1 hESCs relative to siNT hESCs 30 h after silencing. WA09 hESCs were silenced with a mixture of 3 siRNA for FUT1 and 1 NT siRNA. $n = 3$ biological replicates. (D) Left: Representative bright-field images of WA09 colony morphology before silencing and 54 h and 78 h after FUT1 silencing. Right: Quantification of MKI67 and Cyclin D1 mRNA from WT, siNT and siFUT1 hESCs relative to WT hESCs 30 h after silencing. H9.1 and WA09 hESCs were silenced with 3 siRNA for FUT1 and 1 NT siRNA. $n = 2$ biological replicates for each line. (E) Quantification of pluripotent markers (OCT3/4, NANOG, and SOX2) 30 h and 54 h (NANOG) after FUT1 silencing in WT, siNT and siFUT1 hESCs relative to siNT hESCs, as measured by qPCR. H9.1, WA09, WIBR1, and WIBR2 hESCs were silenced with 5 siRNA for FUT1 and 2 NT siRNA. $n = 2$ biological replicates for each line. (F) Top at Left: Quantification of the first heart gene expression, HAND1 54 h and 78 h after FUT1 silencing in WT, siNT and siFUT1 hESCs relative to WT hESCs. Top right at top: Representative bright field images showing WT, siNT, and siFUT1 hESCs 78 h after silencing. Top right in center: Representative images showing nuclei of WA09, siNT, and siFUT1 hESCs stained with DAPI (blue). Top right at bottom: Representative images showing HAND1 protein expression in WT, siNT, and siFUT1 hESCs (green). Bottom: Quantification of the first heart markers, HOPX, SMAD4, and NKX2.5 30 h (NKX2.5), 54 h, and 78 h (SMAD4), and of the second heart markers, AXIN1 and ISL1 30 h after FUT1 silencing in WT, siNT, and siFUT1 hESCs relative to WT hESCs. WA09, WIBR1, and WIBR2 hESCs were silenced with 5 siRNA for FUT1 and 2 NT siRNA. $n = 2$ technical replicates for $n = 3$ lines. (G) Left: Quantification of DE, NE, and PM markers, FOXA2, PAX6, and MSGN1, respectively, 30 h and 54 h (MSGN1) after FUT1 silencing in WT, siNT, and siFUT1 hESCs relative to WT hESCs. Right at top: Representative bright field images showing WA09 hESCs (48 h), hESC-derived PM and siFUT1 hESCs (54 h) after FUT1 silencing. Right in center: Representative images showing nuclei of WA09 hESCs (48 h), hESC-derived PM and siFUT1 hESCs (54 h) stained with DAPI (blue) after FUT1 silencing. Right at bottom: Representative images showing CDX2 protein expression WA09 hESCs (48 h), hESC-derived PM and siFUT1 hESCs (54 h) after FUT1 silencing (green). WA09 hESCs were silenced with 3 siRNA for FUT1 and 1 NT siRNA. $n = 3$ technical replicates. UEA-I, Ulex Europaeus Agglutinin I; PM, paraxial mesoderm; DE, definitive endoderm; NE, neuroectoderm; OM, Opti-Mem transfection medium. Data information: In (A, D, F, G), scale bars represent 100 μm. In (B), data are presented as means ± SDs. Two-tailed Student's t-tests *$p < 0.05$, **$p < 0.038$. In (C–G), data are presented as means ± SDs. Two-tailed Student's t-tests *$p < 0.05$, **$p < 0.01$, ****$p < 0.0001$ or non-significant (NS).

earlier days of differentiation (Fig. EV4H), suggesting that in addition to BMP, certain α1,2-fucosyl structures can impact the signaling axis of receptors other than BMP, compromising a several of DE and NE downstream genes.

When master transcription factors expressed in the lateral plate mesoderm of murine are disrupted or their encoding genes are deleted, the cardiac tube does not form properly (Foley et al, 2019; Zhang et al, 2014). We therefore postulated that even though FUT1 expression is repstored in a late phase of hESC differentiation toward CMs (Fig. 2C), defective LM cells derived from FUT1+ECs could fail to establish normal CMs. To investigate this, hESCs, control ECs, and FUT1+ECs were differentiated into CMs over 20 d using the canonical WNT/β-catenin signaling protocol (Fig. 2A). The three cultures generated areas of contracting cells on day 10, although the hESCs and control cells generated extensive areas of contraction (Movie EV1) relative to FUT1+ECs (Movie EV2). Expression of the CM protein markers cTnT (31.04% ± 15.6%), and α-Myosin (37.3% ± 13.9%) was compromised in CMs derived from FUT1+ ECs, in contrast to their counterparts, where the percentage of the protein markers was high (cTnT 61.7% ± 21.1%, and α-Myosin 69.5% ± 10.4%) (Fig. 5H) as were the cardiac genes TNNT2 and ACTC1 (Fig. 5I), suggesting that overexpression of α1,2-fucosyl glycoconjugates on stem cells constrains commitment to fate decision. Moreover, our findings demonstrate an approach for generating homogenous cells lacking α-fucosyl end groups that could be used in regenerative medicine applications and support the idea that α1,2-fucosyl glycoconjugates are master molecules responsible for fate decisions during human development.

## α1,2-fucosyl GSLs inhibit BMP signaling

The observations so far have linked α1,2-fucosyl GSL alteration during hESC differentiation into cardiac cells and transcription factors associated with the BMP pathway. We therefore tested the hypothesis of whether BMP4 signals could result in a decrease of α-fucosyl end groups and commitment to differentiation after BMP4 treatment. WT hESCs, FUT1+ECs, and control ECs were stimulated for 24 h and 48 h with BMP4, and α-fucosyl end groups were measured using FACS. In comparison to persistent, high expression in FUT1+ECs, time-dependent treatment of WT hESCs resulted in a steady reduction of α-fucosyl end groups, from 20% at 24 h to 30% at 48 h (Fig. EV5A–C). When BMP antagonist LDN193189 was introduced to the medium, there was no reduction in α-fucosyl end groups in hESCs as well as in the control ECs and FUT1+ECs (Figs. 6A, EV5A and 6B,C). However, introducing bFGF, CHIR99021 and Activin A independently to the medium, resulted in a slight reduction of α-fucosyl end groups compared to BMP4, suggesting that BMP4 is likely to be a critical component in the regulation of sugar metabolism during early developmental events. Bulk RNA-seq analysis 48 h after BMP4 treatment also highlighted a cluster of 50 genes implicated in BMP pathway activation (Gunne-Braden et al, 2020; Papadopoulos et al, 2021) (Fig. 6D). Many of the genes in this cluster that are recognized mesoderm markers were upregulated in the control ECs relative to the untreated control ECs. Consistent with our results for FUT1+ECs, we saw that BMP4 stimulation for 48 h was insufficient to amplify the expression of 30 genes to the level observed in BMP4-stimulated control ECs (Fig. 6D), suggesting that α-fucosyl end groups are targets and regulators of the same signaling pathway (Capolupo et al, 2022).

BMP4 binds and activates cell surface receptors, which allows phosphorylation of downstream proteins known as SMAD1/5/8 in a canonical signal transduction pathway (Massagué et al, 2005). To examine whether α-fucosyl end groups regulate a BMP signaling pathway, WT hESCs, FUT1+ECs, and control ECs were stimulated for 24 h with BMP4; then, SMAD 1/5/8 activation was measured in FACS. In hESCs and control ECs we saw that BMP4 stimulation resulted in SMAD 1/5/8 activation, as measured by SMAD 1/5/8

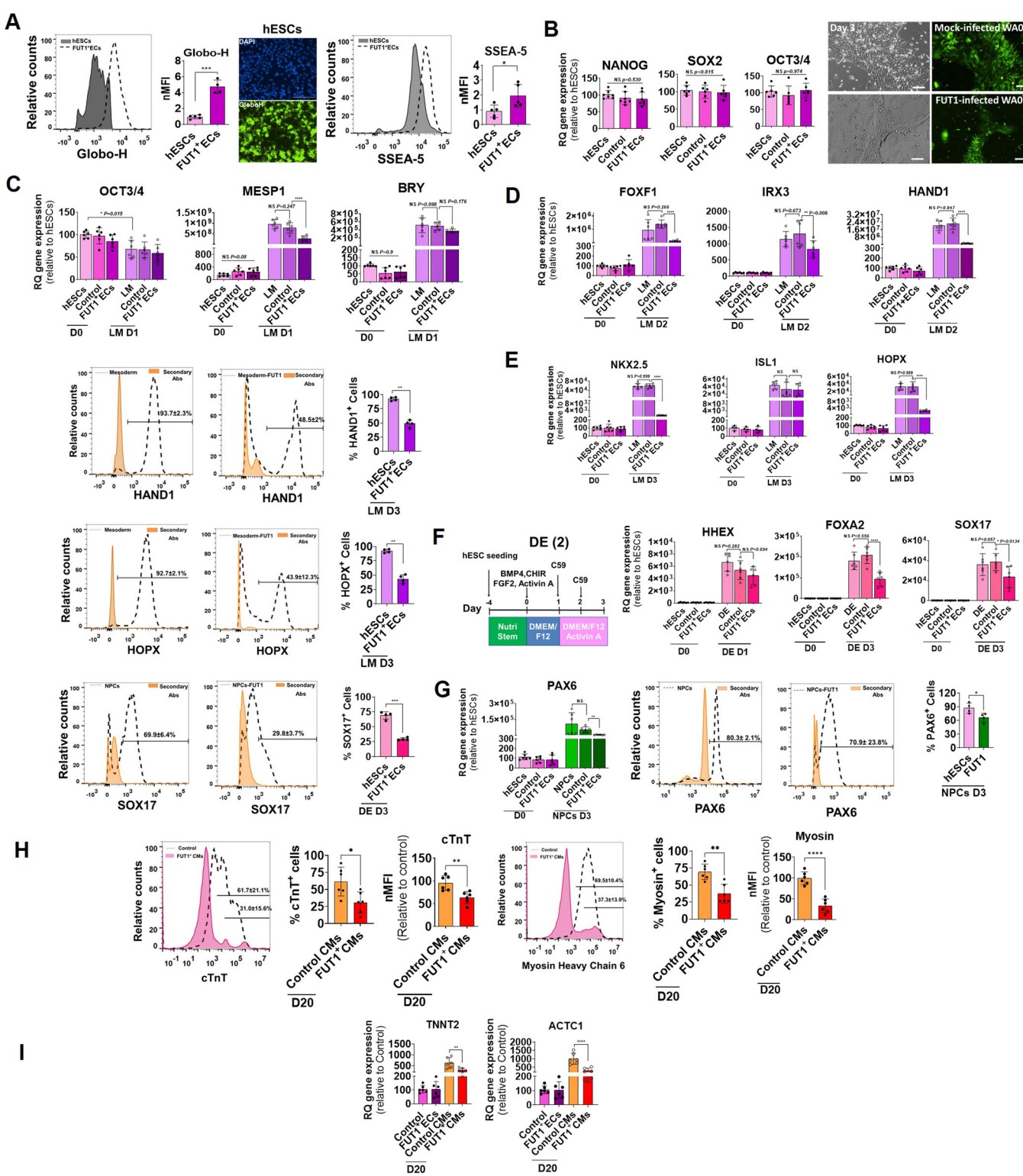

phosphorylation (28.77% ± 6.8% and 32.22% ± 4.9%, respectively), whereas lower phosphorylation (6.07% ± 1.17%) was observed in FUT1+ECs (Fig. 6E). Untreated cells grown in NutriStem revealed inactive SMAD 1/5/8 in WT hESCs (1.23% ± 0.3%), control ECs (0.61% ± 0.36%), and FUT1+ECs (0.47% ± 0.28%) (Fig. 6E), highlighting that constitutive expression of α-fucosyl end groups might interfere with the BMP signaling pathway. To test whether

FUT1 overexpression does interfere with BMP signaling, we assessed SMAD 1/5/8 activity in a time-dependent fashion. In response to continuous stimulation with BMP4, SMAD 1/5/8 phosphorylation in control ECs peaked after 1 h, then increased to a high level after 6 h and subsequently declined to a low level after 12–24 h. When FUT1+ECs were activated with BMP4, a similar single oscillation was seen; the high phosphorylation level was

◀ **Figure 5. A continuous high level of α1,2-fucosyl residues impairs hESC commitment to LM and CMs.**

(A) Left: Representative histograms showing counts and MFI of FUT1 positive hESCs (FUT1+ECs) expressing Globo-H compared to WT hESCs. Center: Representative images showing hESC nuclei (blue) and Globo-H expression (green). Right: Representative histograms showing counts and MFI of FUT1+ECs expressing SSEA-5 compared to hESCs. n = 3 biological replicates. (B) Left: Pluripotent NANOG, SOX2, and OCT3/4 mRNA expression levels in WT hESCs, control hESCs (control) and FUT1+ECs, three passages after mock and FUT1 transfection, respectively. n = 3 clones for each clone n = 2 technical replicates. Right top: Representative bright-field and endogenous fluorescence protein (green) images of control colony morphology after one passage in culture, on day 3. n = 3 clones for each clone n = 2 technical replicates. Right bottom: Representative bright-field and endogenous fluorescence protein (green) images of FUT1+EC colony morphology after one passage in culture, on day 3. n = 2 technical replicates. (C) mRNA expression levels of pluripotent OCT3/4 and mesoderm-specific markers, MESP1 and BRY, in WT hESCs, control, and FUT1+ECs, on day 0 and after differentiation into LM for 1 d. n = 3 clones for each clone n = 2 technical replicates. (D) Top: mRNA expression levels of mesoderm-specific markers, FOXF1, IRX3, and HAND1 in WT hESCs, control, and FUT1+ECs on day 0 and after differentiation into LM for 2 d. n = 3 clones for each clone n = 2 technical replicates. Bottom: Representative histograms showing relative counts of WT hESCs and FUT1+ECs expressing HAND1 protein and quantification of the percent of positive cells for HAND1 in WT hESCs and FUT1+ECs after LM differentiation for 3 d. n = 3 technical replicates. (E) Top: mRNA expression levels of mesoderm-specific markers, NKX2.5, ISL1, and HOPX in WT hESCs, control, and FUT1+ECs on day 0 and after differentiation into LM for 3 d. n = 3 clones for each clone n = 2 technical replicates. Bottom: Representative histograms showing relative counts of WT hESCs and FUT1+ECs expressing HOPX protein and quantification of the percent of positive cells for HOPX in WT hESCs and FUT1+ECs after differentiation into LM for 3 d. Pools of FUT1+ECs were used for FACS. n = 3 technical replicates. (F) Top Left: Schematic showing hESC differentiation into DE over three days by using the protocol of (Loh et al, 2014). Top right: mRNA expression levels of endoderm-specific markers, HHEX, FOXA2, and SOX17 in WT hESCs, control, and FUT1+ECs on day 0 and after differentiation into DE for 3 d, as measured by qPCR. n = 2 clones for each clone n = 3 technical replicates. Bottom: Representative histograms showing relative counts of WT hESCs and FUT1+ECs expressing SOX17 protein and quantification of the fraction of positive cells for SOX17 in WT hESCs and FUT1+ECs after differentiation into DE for 3 d. n = 3 technical replicates. (G) Left: mRNA expression levels of Ectoderm-specific marker, PAX6 in WT hESCs, control, and FUT1+ECs on day 0 and after differentiation into NPCs for 3 d, as measured by qPCR. n = 2 clones for each clone n = 3 technical replicates. Center: Representative histograms showing relative counts of WT hESCs and FUT1+ECs expressing PAX6 protein of the fraction of positive cells for PAX6 in WT hESCs and FUT1+ECs after differentiation into NPCs for 3 d. n = 3 technical replicates. Right: Quantification of positive cells for PAX6 in WT hESCs and FUT1+ECs after differentiation into NPCs for 3 d. n = 2 clones for each clone n = 2 technical replicates. (H) Left: Representative histograms showing relative counts of control and FUT1+ECs expressing cTnT protein after differentiation into CMs for 20 d. n = 2 clones for each clone n = 3 technical replicates. Left at center: Quantification of the fraction of positive cells for cTnT in control and FUT1+ECs after differentiation into CMs for 20 d. Left at right: MFI of control and FUT1+ECs expressing cTnT after differentiation into CMs for 20 d. n = 2 clones for each clone n = 3 technical replicates. Right at left: Representative histograms showing relative counts of control and FUT1+ECs expressing Myosin protein after differentiation into CMs for 20 d. n = 2 clones for each clone n = 3 technical replicates. Right at center: Quantification of the fraction of positive cells for Myosin in control and FUT1+ECs after differentiation into CMs for 20 d. Right at right: MFI of control and FUT1+ECs expressing Myosin after differentiation into CMs for 20 d. n = 2 clones for each clone n = 3 technical replicates. (I) Left: Quantification of cardiac marker, TNNT2 mRNA expression in control and FUT1+ECs on day 0 and after CM differentiation for 20 d, as measured by qPCR. Right: Quantification of cardiac marker, ACTC1 mRNA expression in control and FUT1+ECs on day 0 and after CM differentiation for 20 d, as measured by qPCR. The housekeeping gene GAPDH, was used for normalization. n = 2 clones for each clone n = 3 technical replicates. More than n = 7 clones of control hESCs and FUT1+ECs were generated for overexpression experiments; mRNA expression of n = 3 clones was measured by qPCR. Pools and n = 2 clones of control hESCs and FUT1+ECs originating from WA09-transfected hESCs after sorting were used for imaging and FACS. n = 2 clones of control and FUT1+ECs were differentiated into CMs and measured by qPCR and FACS. Data presented are relative to the values of day 0 WT hESCs. Data information: In (A, B), scale bars represent 100 μm. In (A–F), data are presented as means ± SDs. Two-tailed Student's t-tests *p < 0.05, **p < 0.01, ***p < 0.001, ****p < 0.0001 or non-significant (NS). In (G), data are presented as means ± SDs. Ordinary one-way ANOVA **P < 0.01 or non-significant (NS), and for PAX 6, Two-tailed Student's t-test *p < 0.05. In (H), data are presented as means ± SDs. Two-tailed Student's t-tests *p < 0.05, **p < 0.01, and for Myosin, Ordinary one-way ANOVA **p < 0.01, ****p < 0.0001. In (I), Data are presented as means ± SEM. Two-tailed Student's t-tests *p < 0.05, **p < 0.01, ***p < 0.001, ****p < 0.0001.

reached after 1 h and eventually dropped to lower levels than phosphorylation in control ECs (Fig. EV5B). To see if FUT1 silencing improved BMP signaling, we conducted the SMAD 1/5/8 activity experiment in a time-dependent manner with siNT and siFUT1 hESCs. A wave of phosphorylated SMAD 1/5/8 oscillation was observed in siNT cells for 24 h, whereas siFUT1 cells continuously phosphorylated the SMAD 1/5/8 protein for 24 h (Fig. EV5C). Furthermore, adding LDN193189 to growth medium 30 h after silencing FUT1 resulted in continuous HOPX gene expression as evaluated in siNT and WT hESCs, as well as a reduction in HAND1 relative to siNT cells and WT hESCs (Fig. EV5D), demonstrating that the BMP antagonist lowers HAND1 in siFUT1, unlike siFUT1 in NutriStem medium (Fig. 4F). Overall, these results highlighted the effects of α1,2-fucosyl GSLs on BMP4 signaling and SMAD 1/5/8 activity during hESC differentiation into mesoderm.

## Discussion

By examining the effects of α1,2-fucosyl glycoconjugates' expression in developmental events, we have shown how they are involved in cell-fate decisions.

Our observations support the high amount of α1,2-fucosyl glycoconjugates in pluripotency state and the low level required for proper cell differentiation. We show that constitutive high expression of α1,2-fucosyl glycoconjugates inhibit the BMP

signaling pathway by reducing SMAD 1/5/8 activation, thus possessing a repressive role for the expression of BMP-associated genes in hESCs (Rao et al, 2016), particularly those associated with LM and cardiac cells (Tsaytler et al, 2023).

The FT1, which is encoded by the FUT1 gene catalyzes the synthesis of α1,2-fucosyl glycans, is involved in the biosynthesis of ABO blood group antigens, Lewis antigens, Globo-H and SSEA-5 (Lin et al, 2020; Tang et al, 2011), in both GSLs and glycoproteins (Domino et al, 2001), and the FT2 encoded by FUT2 gene is an important paralog of the FT1. Both enzymes catalyze similar reactions during embryogenesis (Chang et al, 2008); however, whether they have identical functionality during embryogenesis is unclear. Unlike FUT1, we found that FUT2 displays a different expression pattern in early mouse embryos and cardiogenesis. Despite the fact that the two copies of FUT1 and FUT2 are paralogous and expressed from pre-implantation through early organogenesis, both are largely part of clusters of genes that govern the EPI molecular feature in different ways throughout early developmental processes. FUT2 also displays a different expression pattern than FUT1 in hESC-derived LM and cardiac progenitors, as well as it does not catalyze the addition of α1,2-fucosyl residues to glycans after silencing the FUT1 of hESCs. We therefore suggest that FUT1 is a critical GT in pluripotency that must be reduced in order to promote the activity of extracellular pluripotency cues.

The downregulation of the FUT1 gene and the low activity of its encoded GT during the early differentiation of hESC and hiPSC into

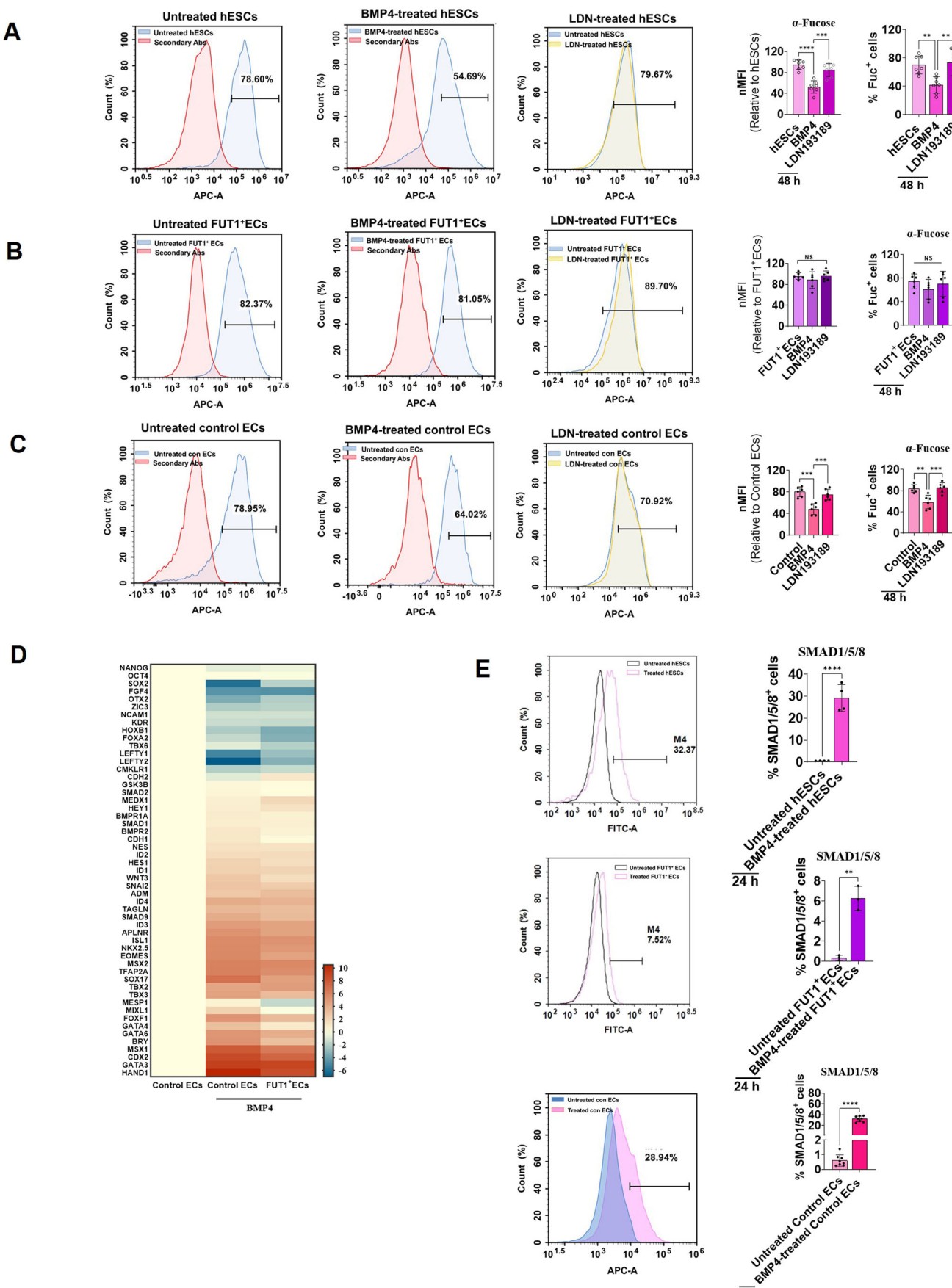

◄ **Figure 6. α-fucosyl glycoconjugates are targets and regulators of the BMP signaling pathway.**

(A) Left at left: Representative histograms showing relative counts of WT hESCs expressing α-fucose residues before BMP4 treatment for 48 h. Left at center: Representative histograms showing relative counts of WT hESCs expressing α-fucose residues after BMP4 treatment for 48 h. Left at right: Representative histograms showing relative counts of WT hESCs expressing α-fucose residues after LDN193189 treatment for 48 h. Right at left: MFI of WT hESCs expressing α-fucose residues before and after BMP4 or LDN193189 treatment for 48 h. Right at right: quantification of the percent of positive WT hESCs expressing α-fucose residues before and after BMP4 or LDN193189 treatment for 48 h. $n = 6$ technical replicates. (B) Left at left: Representative histograms showing relative counts of FUT1[+]ECs expressing α-fucose residues before and after BMP4 treatment for 48 h. Left at center: Representative histograms showing relative counts of FUT1[+]ECs expressing α-fucose residues after BMP4 treatment for 48 h. Left at right: Representative histograms showing relative counts of FUT1[+]ECs expressing α-fucose residues before and after LDN193189 treatment for 48 h. Right: MFI and quantification of the percent of positive FUT1[+]ECs expressing α-fucose residues before and after BMP4 or LDN193189 treatment for 48 h. $n = 6$ technical replicates. (C) Left at left: Representative histograms showing relative counts of control ECs expressing α-fucose residues before and after BMP4 treatment for 48 h. Left at center: Representative histograms showing relative counts of control ECs expressing α-fucose residues after BMP4 treatment for 48 h. Left at right: Representative histograms showing relative counts of control ECs expressing α-fucose residues before and after LDN193189 treatment for 48 h. Right: MFI and quantification of the percent of positive control ECs expressing α-fucose residues before and after BMP4 or LDN193189 treatments for 48 h. $n = 2$ clones were analyzed for each experimental condition. $n = 3$ technical replicates. (D) Heatmap comparing the expression level of 50 genes involved in pluripotency and BMP signaling of control ECs and FUT1[+]ECs after BMP4 (50 ng/ml) treatment for 48 h. The housekeeping gene GAPDH was used for normalization. $n = 3$ technical replicates. (E) Top at left: Representative histograms showing WT hESCs expressing phosphorylated SMAD1/5/8 after BMP4 (50 ng/ml) treatment for 24 h. Top at right: Quantification of the percent of positive WT hESCs expressing activated SMAD1/5/8 after BMP4 (50 ng/ml) treatment for 24 h. Center at left: Representative histograms showing FUT1[+]ECs expressing phosphorylated Smad1/5/8 after BMP4 (50 ng/ml) treatment for 24 h. Center at right: Quantification of the percent of positive control hESCs expressing activated Smad1/5/8 after BMP4 (50 ng/ml) treatment for 24 h. Bottom at left: Representative histograms showing control ECs expressing phosphorylated SMAD1/5/8 after BMP4 (50 ng/ml) treatment for 24 h. Bottom at right: Quantification of the percent of positive control ECs expressing activated Smad1/5/8 after BMP4 (50 ng/ml) treatment for 24 h. $n = 2$ clones of control and FUT1[+]ECs were used for the FACS analysis. $n = 3$ technical replicates. Data information: In (A, B, C), data are presented as means ± SD. Ordinary One-way ANOVA **$p < 0.01$, ***$p < 0.001$, ****$p < 0.0001$ or non-significant (NS). In (D), data are presented as means ± SD. DESeq tests $p < 0.05$. In (E), data are means ± SEM. Two-tailed Student's t-tests *$p < 0.05$, **$p < 0.01$, ***$p < 0.001$, ****$p < 0.0001$.

LM, DE, and NE are governed by extrinsic factors that eventually lead to the expression of differentiation-associated transcription factors. We therefore suggest that cell commitment to fate decisions can, in principle, be mediated by a positive feedback loop, in which an instructive factor, like the instructive BMP4 cue we observed, binds its BMP receptors, BMPR1/2 (Gunne-Braden et al, 2020; Shi and Massagué, 2003) and leads to the production of α1,2-fucosyl-deficient glycoconjugates that activate the factor's receptor and foster differentiation. Nonetheless, different combinations and concentrations of signaling molecules acting directly on EPIs or a few types of progenitors to produce specific cellular responses, do not always result in FUT1 downregulation (Liang et al, 2011), this is likely in order to reduce specific signaling pathways by maintaining the expression of α1,2-fucosyl structures.

Silencing FUT1 causes temporary and reversible loss of self-renewal and stem cell identity. Despite central pluripotent genes are downregulated, and BMP-associated genes are upregulated, the pluripotent markers are recovered 3 d after silencing FUT1 and TFs of cell differentiation decreased to undetectable levels. As well, embryonic cells retain their morphology and proliferation capacity. The findings are consistent with viable and fertile triple knockout homozygous FUT1/FUT2/SEC1 mice (Chen et al, 2023).

FUT1 overexpression primarily compromises mesoderm-associated genes by inhibiting BMP signaling. SMAD phosphorylation has oscillatory behavior (Miller et al, 2019). The minimum duration of BMP4 to trigger loss pluripotency-specific genes and upregulation of primitive streak markers is 30 min (Gunne-Braden et al, 2020), and the levels of phosphorylated SMAD are affected by signal duration and ligand dosage (Miller et al, 2019). Furthermore, phosphorylated SMAD1/5 was discovered to target BMP-master genes in the nucleus of progenitor cells, facilitating lineage development, whereas targeting other BMP genes in the nucleus promoted cell proliferation in stem cells (Genander et al, 2014). Accordingly, the restricted amplitude of phosphorylated SMAD1/5/8 in FUT1[+] ECs and the constitutive phosphorylation in siFUT1 affect downstream gene expression in the BMP signaling pathway. High production of α1,2-fucosyl GSLs leads to decreased activation of SMAD 1/5/8 and reduced ability to translocate to the nucleus and regulate gene transcription.

Therefore, we suggest a bistability model for modulating cell-fate choices through glycan composition (Gunne-Braden et al, 2020; Capolupo et al, 2022), in which a high level of α1,2-fucosyl glycans leads to a decrease in BMP signal transduction and facilitates the pluripotent state via a negative feedback loop, while a low level of α1,2-fucosyl glycans leads to an increase in BMP signaling and facilitates hESC differentiation via a positive feedback loop.

This study sheds light on how different levels of α1,2-fucosyl glycoconjugate expression contribute to maintaining the cell pluripotency state and driving stem cell differentiation by controlling cell signaling. In this regard, extrinsic factors play a role in this, first in changing sugar composition and later in determining cell lineage.

# Methods

**Reagents and tools table**

| Reagent and resource | Source | Identifier |
| --- | --- | --- |
| Antibodies | | |
| Globo-H (VK9) | Thermo Fisher Scientific (mouse, monoclonal) (FACS, IHC-Fr, IF) | 14-9700-82 |
| NANOG | Abcam (rabbit, monoclonal) (FACS) | Ab214549 |
| SSEA3 | Santa Cruz Biotechnology (rat, monoclonal) (FACS) | sc-21703 |

| Reagent and resource | Source | Identifier |
|---|---|---|
| SSEA4 | Santa Cruz Biotechnology (rat, monoclonal) (FACS, IF) | sc-21704 |
| SSEA5 APC | Biolegend (mouse, monoclonal) (FACS) | 355209 |
| OCT3/4 | Santa Cruz Biotechnology (mouse, monoclonal) (FACS) | sc-5279 |
| TRA-1-60 | R&D Systems (mouse, monoclonal) (IF) | MAB4770 |
| SOX2 | GeneTex (mouse, monoclonal) (FACS) | GTX627404 |
| Brachyury | Santa Cruz Biotechnology (goat, polyclonal) (IHC-Fr) | sc-17743 |
| FT1 (human) | Thermo Fisher Scientific (rabbit, polyclonal) (FACS) | PA5-13515 |
| FT1 (mouse) | LifeSpan BioSciences (rabbit polyclonal) (IHC-Fr) | LS-C407806 |
| MIXL1 | Mercury (rabbit, polyclonal) (IHC-Fr) | ABS232 |
| CDX2 AF 488 | Abcam (rabbit, polyclonal) (IHC-Fr) | ab195007 |
| HAND1 | R&D Systems (goat, polyclonal) (FACS) | AF3168 |
| HOPX | Abcam (rabbit, monoclonal) (FACS) | Ab106251 |
| NKX2.5 | Santa Cruz Biotechnology (mouse, monoclonal) (FACS) | sc-376565 |
| PAX6 | Thermo Fisher Scientific (rabbit, polyclonal) (FACS, IF) | 42-6600 |
| OTX2 AF488 | R&D Systems (mouse, monoclonal) (IHC-Fr) | IC979G |
| SOX17 | Abcam (mouse, monoclonal) (FACS) | Ab84990 |
| SIRPA APC | Biolegend (mouse, monoclonal) (FACS, IF) | 323810 |
| cTnT | Abcam (rabbit, monoclonal) (FACS, IF) | Ab8295 |
| Cardiac myosin | GeneTex (mouse, monoclonal) (FACS, IF) | GTX20015 |
| Sarcomeric-α actinin | Sigma-Aldrich-Merck (mouse, monoclonal) (IF) | A7811 |
| Phospho-SMAD1/5/9 | Cell Signaling Technology (rabbit, monoclonal) (FACS, Western blotting) | 13820 |
| α-Fetoprotein (αFP) | R&D Systems (mouse, monoclonal) (FACS) | MAB1368 |
| HRP-β-Actin | Santa Cruz Biotechnology (mouse, nonoclonal) (Western blotting) | SC-47778 |
| Chemicals, Proteins, and Factors | | |
| FGF-basic | PeproTech (Recombinant human protein) | 100-18B |
| BMP4 | PeproTech (Recombinant human protein) | 120-05 |
| LDN193189 | BioGems | 1062443 |
| CHIR 99021 | Tocris | 4423 |
| Wnt-3a | PeproTech (Recombinant murine protein) | 315-20 |
| Activin-A | PeproTech (Recombinant human/murine/rat protein) | 120-14 |
| IWP-2 | Tocris (WNT inhibitor) | 3533 |
| Dorsomorphin | Tocris (BMP type I receptor inhibitor) | 3093 |
| A-83-01 | BioGems (TGF-β type I receptor inhibitor) | 9094360 |
| Retinoic acid | Sigma-Aldrich-Merck | R2625 |
| ROCK inhibitor Y-27632 | Cayman Chemical | 10005583 |
| Phalloidin AF488 | Thermo Fisher Scientific (IF for F-actin staining) | A12379 |
| UEA-I biotinylated | Vector Laboratories | B-1065-2 |
| siRNA sequences | | |

| Target gene | Duplex sequence | IDT duplex name |
|---|---|---|
| *HPRT* | 5'-GCCAGACUUUGUUGGAUUGGAAA-3'<br>5'-AAUUUCAAAUCCAACAAAGUCUGGCUU-3' | HPRT-S1 DS |
| *NC-1* | 5'-CGUUAAUCGCGUAUAAUACGUGUA-3'<br>5'-AUACGCGUAUUAUACGCGAUUAACGAC-3' | NC1 control duplex |
| *NC-2* | 5'-CUUCCUCUCUUUCUCUCCCUUGUGA-3'<br>5'-UCACAAGGGAGAGAAAGAGAGGAAGGA-3' | NC2 |

| Reagent and resource | Source | Identifier |
|---|---|---|
| *FUT1* | 5'-GGGAGUUACAGUUACAAUUGUUACA-3'<br>5'-UGUAACAAUUGUAACUGUAACUCCCUG-3' | NM_000148 duplex1 |
| *FUT1* | 5'-GGAAGACAGGUUGGCUAAUUUCCTG-3'<br>5'-CAGGAAAUUAGCCAACCUGUCUUCCCU-3' | NM_000148 duplex2 |
| *FUT1* | 5'-UCCAGAUAACUAAGGUGAAGAAUCT-3'<br>5'-AGAUUCUUCACCUUAGUUAUCUGGAUU-3' | NM_000148 duplex3 |
| *FUT1* | 5'-CCACUCUGGACAUUGGCUAAGCCTT-3'<br>5'-AAGGCUUAGCCAAUGUCCAGAGUGGAG-3' | NM_000148 13.2 |
| *FUT1* | 5'-UUGAGAGAUCCUUUCCUGAAGCUCT-3'<br>5'-AGAGCUUCAGGAAAGGAUCUCUCAAGU-3' | NM_000148 13.3 |
| Recombinant DNA (Plasmids) | | |
| pHAGE2-FullEF1A-DsRedExpress-IRES-ZsGreen | This paper | |
| pHAGE2-FullEF1A-FUT1Express-IRES-ZsGreen | This paper | |
| Critical Commercial Assays | | |
| Lipofectamine RNAiMAX transfection reagent | Thermo Fisher Scientific | 13778075 |
| Deposited Data | | |
| RNA-Seq | Li et al (2014)<br>https://www.ncbi.nlm.nih.gov/geo/query/acc.cgi?acc=GSE51483 | GEO: GDS5003 |
| Sc-RNA-Seq | Peng et al (2019)<br>https://www.ncbi.nlm.nih.gov/geo/query/acc.cgi?acc=GSE120963 | GEO: GSE120963 |
| RNA-Seq | Gu et al (2014)<br>https://www.ncbi.nlm.nih.gov/geo/query/acc.cgi?acc=GSE48257 | GEO: GSE48257 |
| RNA-Seq | Loh et al (2016)<br>https://www.ncbi.nlm.nih.gov/geo/query/acc.cgi?acc=GSE85066 | GEO: GSE85066 |
| RNA-Seq | Loh et al (2014)<br>https://www.ncbi.nlm.nih.gov/geo/query/acc.cgi?acc=GSE52657 | GEO: GSE52657 |
| RNA-Seq | This paper<br>https://www.ncbi.nlm.nih.gov/sra/PRJNA1112109 | PRJNA1112109 |
| Raw data | This paper<br>https://www.ebi.ac.uk/biostudies/studies/S-BSST1426?key=22235c8f-e525-455c-8b88-053d58da8348 | Biostudies, accession number S-BSST1426 |
| Cell lines | | |
| WA09 hESC | WiCell | RRID:CVCL_9773 |
| WA09.1 hESC | WiCell | RRID:CVCL_C811 |
| WIBR1 hESC | Whitehead Institute | RRID:CVCL_9765 |
| WIBR2 hESC | Whitehead Institute | RRID:CVCL_9766 |
| BJ hiPSC | BGU | RRID:CVCL_A4YN |
| EMF hiPSC | BGU | RRID:CVCL_A4ZP |
| OME hiPSC | BGU | RRID:CVCL_A4YM |
| WA09 hESC FUT1 | This paper | Overexpression of FUT1 |
| WA09 hESC Control | This paper | Mock-transfected hESCs |
| Software and Algorithms | | |
| 2D, 3D plots | Peng G, Suo S, Cui G, Yu F, Wang R, Chen J, Chen S, Liu Z, Chen G, Qian Y, et al (2019) | http://egastrulation.sibcb.ac.cn |
| FUT1 sequences (primers) | | |
| | **Sequences** | |
| *Forward* | 5'-GCAGCGGCCGCCTAGCCTGCCCTGGGTGAA-3' | |
| *Reverse* | 5'-TTAGGATCCTTCTAGAACTGCCTGCCAGC-3' | |

| Reagent and resource | Source | | Identifier |
|---|---|---|---|
| TaqMan assay data for qPCR | | | |
| Standard gene name | Full gene name | Gene product | TaqMan gene expression assay ID |
| GAPDH | Glyceraldehyde-3-phosphate dehydrogenase | GAPDH | Hs99999905_m1 |
| ACTB | Actin beta | Actin beta | Hs99999903_m1 |
| HPRT | hypoxanthine phosphoribosyltransferase 1 | HPRT | Hs99999909_m1 |
| POU5F1 | POU class 5 homeobox 1 | OCT4 | Hs04260367_gH |
| NANOG | Nanog homeobox | NANOG | Hs04399610_g1 |
| SOX2 | SRY-box Transcription Factor 2 | SOX2 | Hs01053049_s1 |
| SOX1 | SRY-Box Transcription Factor 1 | SOX1 | Hs01057642_s1 |
| GATA4 | GATA binding protein 4 | GATA4 | Hs00171403_m1 |
| NKX2.5 | NK2 homeobox 5 | NKX2.5 | Hs00231763_m1 |
| ISL1 | ISL LIM homeobox 1 | ISL1 | Hs00158126_m1 |
| TNNT2 | Troponin T cardiac type 2 | Troponin T | Hs00943911_m1 |
| ACTC1 | Actin Alpha Cardiac Muscle 1 | Alpha-Cardiac Actin | Hs01109515_m1 |
| FUT1 | fucosyltransferase 1 (galactoside 2-alpha-L-fucosyltransferase) | FUT1 | Hs01379722_m1 |
| FUT1 | Fucosyltransferase 1 (galactoside 2-alpha-L-fucosyltransferase) | FUT1 plasmid | Hs00355741_m1 |
| FUT2 | Fucosyltransferase 2 (galactoside 2-alpha-L-fucosyltransferase 2) | FUT2 | Hs00704693_s1 |
| FUT3 | Blood group Lewis alpha-4-fucosyltransferase | FUT3 | Hs00356857_m1 |
| B3GALT2 | UDP-Gal:betaGlcNAc beta 1,3-galactosyltransferase; polypeptide 2 | B3GALT2 | Hs00705203_s1 |
| B3GALNT1 | Beta-1,3-N-acetylgalactosaminyltransferase 1 (globoside blood group) | B3GALNT1 | Hs00364202_s1 |
| B3GNT5 | UDP-GlcNAc:betaGal beta-1,3-N-acetylglucosaminyltransferase 5 | B3GNT5 | Hs00908059_m1 |
| UGCG | UDP-glucose ceramide glucosyltransferase | Glucosylceramide synthase | Hs00234293_m1 |
| ST3GAL1 | ST3 beta-galactoside alpha-2,3-sialyltransferase 1 | ST3GAL1 | Hs00161688_m1 |
| HOPX | HOP homeobox | HOPX | Hs04188695_m1 |
| FOXF1 | Forkhead box F1 | FOXF1 | Hs00230962_m1 |
| IRX3 | Iroquois homeobox 3 | IRX3 | Hs01124217_g1 |
| SMAD4 | SMAD family member 4 | SMAD4 | Hs00929647_m1 |
| PAX6 | Paired box 6 | PAX6 | Hs00240871_m1 |
| FOXA2 | Forkhead box A2 | FOXA2 | Hs00232764_m1 |
| SOX17 | SRY-box transcription factor 17 | SOX17 | Hs00751752_s1 |
| AXIN1 | Axin 1 | AXIN1 | Hs00959582_m1 |
| MSGN1 | Mesogenin 1 | MSGN1 | Hs03405514_s1 |
| HHEX | Hematopoietically-expressed homeobox | HHEX | Hs01074519_m1 |
| T | T Brachyury transcription factor | BRY | Hs00610080_m1 |
| MESP1 | Mesoderm posterior protein 1 | MESP1 | Hs00251489_m1 |
| OTX2 | Orthodenticle homeobox 2 | OTX2 | Hs00222238_m1 |
| VIM | Vimentin | Vimentin | Hs00185584_m1 |
| KI67 | Marker of proliferation Ki67 | MKI76 | Hs01032443_m1 |
| Cyclin D1 | B-cell lymphoma 1 protein | CCND1 | Hs00277039_m1 |

## Experimental model

All experiments were performed using either hESC lines WA09 (H9) (WiCell; RRID:CVCL_9773) and WA09.1 (H9.1) (WiCell; RRID:CVCL_C811) WiCell, Madison, originally derived by the Thomson Lab (Amit et al, 2000; Thomson et al, 1998) and WIBR1 (Whitehead Institute; RRID:CVCL_9765) and WIBR2 (Whitehead Institute; RRID:CVCL_9766), originally derived by the Jaenisch Lab (Lengner et al, 2010) or iPSC lines BJ (BGU; RRID:CV-CL_A4YN), EMF (BGU: RRID:CVCL_A4ZP), and OME (BGU; RRID:CVCL_A4YM), originally derived by the Ofir Lab (Naaman et al, 2018). Genome-edited clonal lines were generated in this study from WA09, WA09.1, WIBR1, and WIBR2 cell lines, and were routinely cultured in serum-free, feeder-free conditions on growth factor reduced Matrigel-coated plates (BD Biosciences). Cells were fed daily using chemically defined medium (NutriStem hESC XF medium, Sartorius) and 1% penicillin streptomycin (P/S), incubated at 37 °C with 5% $CO_2$ and passaged every 3–4 d. A gentle dissociation EDTA buffer (Sartorius) was used for passaging. The quality of hESC lines was routinely assessed by qPCR and flow cytometry of multiple pluripotent markers. Cells were routinely screened for mycoplasma (LiLiF Diagnostics).

Lentivirus was generated in HEK293T cells (ATCC Cat# CRL-3216; RRID:CVCL_0063), which were maintained in DMEM (GIBCO) with 10% fetal bovine serum (FBS) and 1% P/S (GIBCO), at 37 °C with 5% $CO_2$ and passaged once every 4–5 d by dissociation using Trypsin-EDTA (Sartorius).

Animal procedures were conducted as approved by the local authorities (Ben-Gurion University, Beer-Sheva) under the license number IL-84-10-2018(C). Mouse embryos used in this study were dissected from C57BL/6J genetic background according to standard protocol (Pryor et al, 2012).

## Differentiation of hESCs into EBs

Matrigel-adherent hESCs, at a confluence of 80%, were dispersed into small clumps using Versene (GIBCO). The clumps were cultivated in suspension on low-attachment plates in an EB formation medium [79% DMEM, 20% FBS, 1% nonessential amino acids, 1% P/S, 1 mM L-glutamine, 0.1 mM β-mercaptoethanol, and 5 μM ROCK inhibitor Y-27632; (R&D Systems)] for 5 d.

## Differentiation of hESCs into CMs using Wnt signaling and inhibition protocol

Cardiomyocytes were generated as previously reported (Lian et al, 2013), with minor modifications. Briefly, hESCs were maintained on Matrigel in a NutriStem medium and dissociated with Accutase (Thermo Fisher Scientific). Then, the cells were seeded on a Matrigel-coated cell culture plate at a density of $2 \times 10^5$ cells per $cm^2$ in NutriStem supplemented with 5 μM Y-27632 for 24 h (day −4). When the plate was fully confluent, after 4 d (day 0), the cells were supplemented with 5 μM CHIR99021 (Tocris) in RPMI/B27 (GIBCO). After 24 h (day 1), the medium was changed to RPMI/B27-insulin. After 48 h, 5 μM IWP2 (Tocris) was added for an additional 48 h; then it was removed, and the medium was changed to RPMI/B27-insulin (GIBCO) for another 48 h (day 7). The RPMI/B27 medium was changed every 48 h.

Spontaneous cell contractions first appeared on day 10 and on subsequent days.

## Differentiation of hESCs into LM in chemically defined conditions

For the differentiation of hESCs into LM the protocol of (Loh et al, 2016) was used. Briefly, 48 h post-seeding hESCs at a density of $2.5 \times 10^4$ cells per $cm^2$ on Matrigel in a NutriStem medium were differentiated by cultivation with 30 ng/ml Activin A (PeproTech), 40 ng/ml BMP4 (PeproTech), 6 μM CHIR99021, and 20 ng/ml FGF2 (PeproTech) (day 1). After 24 h, the cells were supplemented with 1 μM A-83-01, 30 ng/ml BMP4, 20 ng/ml FGF2, and 1 μM C59 for the next 24 h (day 2) and then treated with 30 ng/ml BMP4, 1 μM A-83-01, and 20 ng/ml FGF2 for an extra 24 h (day 3).

## Differentiation of hESCs into DE in chemically defined conditions

Two protocols were achieved definitive endoderm differentiation. The first protocol was that previously reported for hepatocytes (Hinton et al, 2010): at a cell confluence of ~100%, the hESCs were supplied with 100 ng/ml Activin A in RPMI for 4 d. The medium was supplemented with 25 ng/ml Wnt-3a (PeproTech) for 24 h and then replaced by a 0.2% FBS/RPMI medium supplemented with 100 ng/ml Activin A for an additional 48 h. After 48 h, the medium was replaced by 2% FBS/RPMI supplemented with 100 ng/ml Activin-A. Finally, the cells were supplied with 2% FBS/RPMI without supplementary growth factors for another 4 d. The second protocol was adopted from Loh et al (Loh et al, 2014), with slight modifications. At a cell confluence of 50%, the hESCs were exposed to a DMEM-F12 medium (Sartorius) supplemented with 20 ng/ml FGF2, 60 ng/ml Activin A, 3 μM CHIR99021, 10 ng/ml BMP4, and 1% P/S for 24 h, then medium was changed and supplemented with 20 ng/ml FGF2, 60 ng/ml Activin A, 1 μM C59, and 250 nM LDN193189 (PeproTech) for an additional 48 h.

## Differentiation of hESCs into NPCs in chemically defined conditions

Differentiation of hESCs into NPCs was performed as previously described (Qu et al, 2014), with some modifications. Dissociated cells were cultured on Matrigel-coated plates in a NutriStem medium to achieve a cell confluence of 80%. The cells were then differentiated in DMEM/F12 supplemented with 1% $N_2$ (Stem Cell Technologies), 1 μM dorsomorphin (TOCRIS), 1 mM L-glutamine, 1% P/S, and 0.2% heparin. After 72 h, 100 nM retinoic acid (RA, PeproTech) was added to the medium for an additional 72 h.

## Generating hESC lines in which FUT1 has been temporarily knockdown (KD) using siRNA

hESCs (WA09, WIBR1, and WIBR2) were seeded on a Matrigel-coated plate at a cell density of 85,000 cells per $cm^2$ in a culture of NutriStem medium. Twenty-four hours post-seeding, the cells were incubated with the transfection media [10% (v/v) Opti-MEM (OM) with lipofectamine RNAiMAX (Thermo Fisher Scientific) and

relevant siRNA (IDT)] diluted in a fresh NutriStem medium. At predefined time points, transcripts of silenced cells and protein expression were detected using qRT-PCR and flow cytometry. Negative control groups either without siRNA (control) or with non-targeting siRNA (siNT) were used. Details on the siRNA sequences are provided in the reagents and tools table.

## Generating constitutive FUT1 hESC lines using lentiviral packaging plasmid

WA09 hESCs were used to generate clonal lines of FUT1+ECs; briefly, primers designed to clone a wild-type DNA of the FUT1 coding sequence from extracted cDNA of WA09 cells using PCR amplification and amplicon were cut by restriction site enzymes NotI and BamHI (Thermo Fisher Scientific). Details on the Primer sequences are provided in the reagents and tools table. The PCR product was cloned into lentiviral vector pHAGE2-FullEF1a-DsRedExpress-IRES-ZsGreen, provided by Prof. Roi Gazit (Ben-Gurion University of the Negev, Israel) (Keinan et al, 2021), by replacing the DsReD sequence using the corresponding restriction enzymes. The original lentiviral vector was used as a control. To produce virions, a lentiviral vector expressing the FUT1 cDNA was packaged in HEK293T cells using the packaging plasmids pMD2.G and psPAX2 (Addgene). The virions were collected 48 h after transfection. After treatment with 6 μg/ml polybrene (Calbiochem), hESCs were transfected for 48 h. The cells were sorted according to their ZsGreen expression by a FACSAria instrument (BD Biosciences, San Diego, CA) to ensure FUT1 expression and then deposited into Matrigel-coated 6 wells and grown in NutriStem with CloneR supplement (1:10, STEMCELL Technologies) for 2 days. CloneR was replaced by fresh medium, which then was changed daily until green clones were seen. Clones were detached mechanically by tipping using the EVOS microscope (Thermo Fisher Scientific) and cultured into new Matrigel-coated 6 wells and grown in NutriStem.

## Quantitative real-time PCR (qPCR)

RNA was extracted using the PureLink RNA Mini Kit (Invitrogen) according to the manufacturer's instructions. For cDNA preparation, the high-capacity cDNA Reverse Transcription Kit (Applied Biosystems) was used according to the manufacturer's instructions. cDNA was prepared from 0.5 to 2 μg mRNA in RNase-free conditions. RNA purity and quantity were assessed using NanoDrop (A260/A280 1.5-2 was considered suitable for further analysis). Gene expression analysis was performed using TaqMan gene expression assays (Applied Biosystems). Reactions were run on a StepOnePlus applied detection system (Applied Biosystems), and PCR conditions consisted of 40 cycles of 95 °C for 10 s, followed by 60 °C for 30 s. The housekeeping genes ACTB and GAPDH were used as normalization controls, where relative gene expression of target genes was calculated by the delta Ct method. TaqMan assay data are detailed in reagents and tools table.

## FACS

Cells were dissociated into a single-cell suspension and fixed using the eBioscience Intracellular Fixation & Permeabilization buffer set,

according to the manufacturer's instructions (Thermo Fisher Scientific). The cells were stained with relevant antibodies suspended in a FACS buffer (PBS with 2% FBS), and cell analysis was performed using a FACS Canto machine (BD Biosciences), utilizing CellQuest Pro software (BD Biosciences). The primary antibodies used were OCT3/4 (Santa Cruz Biotechnology; sc-5279), NANOG (Abcam; ab214549), human FT1 (Thermo Fischer Scientific; PA5-13515), Globo-H (Thermo Fischer Scientific; 14-9700-82), SSEA-3 (Santa Cruz Antibodies; sc-21703), SSEA-4 (Santa Cruz Biotechnology; sc-21704), SSEA-5 APC (BioLegend; 355209), Phospho-SMAD 1/5/9 (Cell Signaling Technology; 13820), NKX2.5 (R&D Systems; AF2444), cTnT (Abcam; ab209813), SIRPA APC (Biolegends; 323810), Cardiac myosin heavy chain 6 (GenTex; GTX20015), HAND1 (R&D Systems; AF3168), SOX17 (Abcam; ab84990), HOPX (Abcam; ab106251), AFP (R&D Systems; MAB1368) and biotinylated Ulex Europaeus Agglutinin I (UEA I) (Vector Laboratories; B-1065-2). The secondary antibodies (Jackson ImmunoResearch) were donkey anti-mouse Alexa Fluor 488 IgG (715-545-151), donkey anti-mouse Alexa Fluor 647 IgG (715-605-151), donkey anti-rabbit Alexa Fluor 488 IgG (711-545-152), donkey anti-goat Alexa Fluor 488 IgG (705-546-147), donkey anti-rabbit Alexa Fluor 647 IgG (711-605-152), and Streptavidin Alexa Fluor 488 (016-540-084). The normalized mean fluorescence intensity (nMFI) was calculated for each representative MFI histogram.

## Immunostaining and confocal microscopy

Cells were fixed with 2% formaldehyde for 2 min and then with 4% paraformaldehyde (PFA) for 7 min, permeabilized with 0.2% (v/v) Triton-X 100 for 5 min, blocked for 2 h in 3% BSA (Millipore) and incubated overnight with primary antibodies: mouse FT1 (LSBio; LS-C407806), TRA-1-60 (R&D Systems; MAB4770), α-actinin (Sigma-Aldrich-Merck; A7811), cTnT (Abcam; ab209813), OTX2 Alexa Fluor 488 (R&D Systems; IC1979G) and PAX6 (Thermo Fischer Scientific; 42-6600) followed by 4 h incubation with the following secondary antibodies: donkey anti-mouse Alexa Fluor 488 antibodies, phalloidin Alexa Fluor 546 (Life Technologies; A22283) used for F-actin staining, and VECTASHIELD Mounting Medium with DAPI (Vector Laboratories; H-1200) for nuclei detection. Imaging was performed with a Nikon C1si laser scanning confocal microscope.

## Immunofluorescence labeling and data acquisition

Mouse embryos (E5.25–8.75) were harvested, washed in PBS, and fixed in PFA (4%) overnight at 4 °C. After three washes in PBS containing 0.05% Tween (PBS-T), the samples were permeabilized using 0.5% Triton X-100 in PBS, washed three times in PBS-T, blocked with a 2% rat serum and 3% BSA for 1 h at room temperature, and incubated overnight at 4 °C in a blocking solution with the following primary antibodies: Globo-H, BRY (Santa Cruz Biotechnology; sc17743), MIXL1 (Mercury; ABS232), mouse FT1 (LSBio; LS-C407806), and CDX2 Alexa Fluor 488 (Abcam; ab195007). After four washes in PBS-T, unlabeled samples were incubated with secondary antibodies Alexa Fluor 488 or 647 in a blocking solution for 1 h at room temperature, washed three times in PBS-T, and mounted in VECTASHIELD with DAPI. The samples were imaged after 1–3 d using a Nikon C1si laser scanning confocal microscope.

## BMP signaling

For BMP signaling, WA09, FUT1+ECs (3 clones), and control (3 clones) hESCs were treated with BMP4 (50 ng/ml) or LDN193189 (250 μM) in NutriStem for 24 and 48 h before cells were washed with PBS and collected to measure α-fucose and SMAD 1/5/8 phosphorylation in FACS. The negative control was untreated cells from each line.

## Western blotting

Cells were lysed in RIPA lysis buffer containing phosphatase and protease inhibitors (Sigma-Merck), followed by four cycles of freezing and thawing to disrupt cellular membranes and extract proteins. Equal amounts of proteins were loaded onto 12% SDS-PAGE gels, then electro-transferred onto nitrocellulose membranes. After blocking with 4% non-fat dry milk, the membranes were incubated overnight at 4 °C with SMAD1/5/8 antibodies (Cell signaling). After washing, the membranes were incubated with HRP-labeled anti-rabbit IgG and β-actin (Jackson antibodies), followed by washing steps to remove unbound antibodies. Finally, the membranes were incubated with developing reagent for 1 min, and the visualized bands were detected using an ECL detection system. The band intensities were estimated in Image J software. The values were observed by normalizing actin bands to those at time zero, normalizing pSMAD bands to pSMAD bands at time zero, and then dividing the normalized pSMAD values by normalized actin values.

## RNA-Seq

FUT1+ECs (3 clones) and control (3 clones) hESCs were treated with BMP4 (50 ng/ml, PeproTech) in NutriStem for 48 h before cells were washed with PBS, and total RNA was extracted using the PureLink RNA Mini Kit (Thermo Fischer Scientific) as per the manufacturer's instructions. RNA quality and quantity were tested using a 2100 Bioanalyzer instrument (Agilent). Samples were sequenced using Illumina technology, yielding single end reads. Two rounds of sequencing were performed. For each sample, reads were quality and adaptor trimmed using Trim Galore. Clean reads were aligned to the human genome (GRCh38.105 from Ensembl version) using STAR, and the read counts per gene per sample were estimated using RSEM. Quality assessment (QA) of the process was performed using FastQC and MultiQC. Gene annotation was retrieved from Ensmbl BioMart. Links to external databases were retrieved using the AnnotationHub R package. Human genes with DNA-binding transcription factor activity (GO:0003700) were retrieved from Ensembl BioMart (https://www.ncbi.nlm.nih.gov/sra/PRJNA1112109).

## RNA-Seq data analysis

Microarray and GEO-seq data were obtained from four studies, as mentioned above. Heat map data were analyzed by using a min-max scale normalization. Single-cell RNA-seq and GEO-seq were analyzed by using (http://egastrulation.sibcb.ac.cn). Statistical analysis for differential expression was performed using DESeq2. Genes with FDR adjusted *p*-value <0.05 in any of the comparisons were regarded as differentially expressed. Fold change in a heatmap is shown in logarithmic scale, with a negative sign denoting downregulation.

## Statistical analysis

Two-tailed student's t-test with a Bonferroni correction and repeated-measures one-way ANOVA with Tukey's post-hoc tests were used. Results were expressed as means ± SEM. Values of $P < 0.05$ indicate significance.

## Data availability

The source data of this paper are collected in the following database record: Biostudies: S-BSST1426 https://www.ebi.ac.uk/biostudies/studies/S-BSST1426?key=22235c8f-e525-455c-8b88-053d58da8348.
The datasets utilized and produced in this study are available in the following databases: RNA-Seq: Gene Expression Omnibus GDS5003 https://www.ncbi.nlm.nih.gov/geo/query/acc.cgi?acc=GSE51483. Sc-RNA-Seq: Gene Expression Omnibus GSE120963. RNA-Seq: Gene Expression Omnibus GSE48257. RNA-Seq: Gene Expression Omnibus GSE85066. RNA-Seq: Gene Expression Omnibus GSE52657. RNA-Seq: Gene Expression Omnibus PRJNA1112109.

The source data of this paper are collected in the following database record: biostudies:S-SCDT-10_1038-S44319-024-00243-1.

## Peer review information

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

## Acknowledgements

We thank Dr. Rivka Ofir from Ben-Gurion University of the Negev for the three lines of iPSCs, Prof. Roi Gazit from Ben-Gurion University of the Negev for the pHAGE2-FullEF1a-DsRedExpress-IRES-ZsGreen, and Prof. Jacob Hanna from the Weizmann Institute of Science for the two lines of hESCs, WIBR1, and WIBR2. We thank Mrs.Tatiana Rabinski for technical assistance and Dr. Vered Chalifa Caspi, head of the Bioinformatics Core Facility at Ben-Gurion University of the Negev for GEO-seq and RNA-seq data analysis. We thank Mrs. Sharon Amlani for illustrating the graphical abstract. Funding and resources were provided by the Jordan Baruch Stem Cell Research Fund at the RMSC Research Center. BR was supported by grants from the Israel Science Foundation (ISF) (1416/19 and 2841/19). DHN was supported by a doctoral fellowship from the Kreitman School of Advanced Graduate Studies at Ben-Gurion University of the Negev. Prof. Smadar Cohen is the incumbent of the Claire and Harold Oshry Professorial Chair in Biotechnology.

## Author contributions

**Saray Chen**: Conceptualization; Formal analysis; Investigation; Methodology. **Dana Hayoun-Neeman**: Conceptualization; Formal analysis; Investigation; Methodology. **Michal Nagar**: Formal analysis; Investigation. **Sapir Pinyan**: Formal analysis; Validation. **Limor Hadad**: Data curation; Investigation; Methodology. **Liat Yaacobov**: Data curation; Investigation. **Lilach Alon**: Investigation. **Liraz Efrat Shachar**: Formal analysis; Investigation; Methodology. **Tair Swissa**: Formal analysis; Investigation; Methodology. **Olga Kryukov**: Resources. **Orly Gershoni-Yahalom**: Data curation. **Benyamin Rosental**: Formal analysis; Visualization. **Smadar Cohen**: Methodology. **Rachel G Lichtenstein**: Conceptualization; Supervision; Investigation; Methodology; Writing—original draft; Writing—review and editing.

Source data underlying figure panels in this paper may have individual authorship assigned. Where available, figure panel/source data authorship is listed in the following database record: biostudies:S-SCDT-10_1038-S44319-024-00243-1.

## Disclosure and competing interests statement

The authors declare no competing interests.

# Expanded View Figures

**Figure EV1. Co-expression of FUT1 with pluripotency and primary germ layer markers in early embryonic development.**

(A) Left: Representative in silico 3D transcriptomic analysis showing co-expression of FUT1 with pluripotent markers (NANOG, SOX2, and OCT3/4) on E6.0 in sections #351–370 from a sequential series of sections. Right: 2D corn plots showing FUT1 and pluripotent gene expression on E6.0. Abbreviations represent A, anterior; P, posterior; L, left; R, right; Pr, proximal; D, distal. (B) 2D corn plots showing the expression of ectoderm markers, OTX2, PAX6, ZIC1, SOX2, SOX1, HOXC8, PHOX2b, OLIG2, PAX7, HOXB9, and DBX1 on E7.5 embryos. (C) 2D corn plots showing the expression of DE markers, HHEX, FOXA2, SOX17, and FOXA1 on E7.5. (D) 2D corn plots showing the spatiotemporal expression mesoderm genes, BRY (T gene) and MIXL1 on E7.0–E7.5; HOPX, NKX2.5, ISL1, HAND1, IRX3, MESP2, FOXC1, FOXF1, and CDX2 on E7.5. 2D and 3D gene expression analysis was obtained by using the database of mouse gastrulation on E5.5–E7.5 (http://egastrulation.sibcb.ac.cn).

▶

 

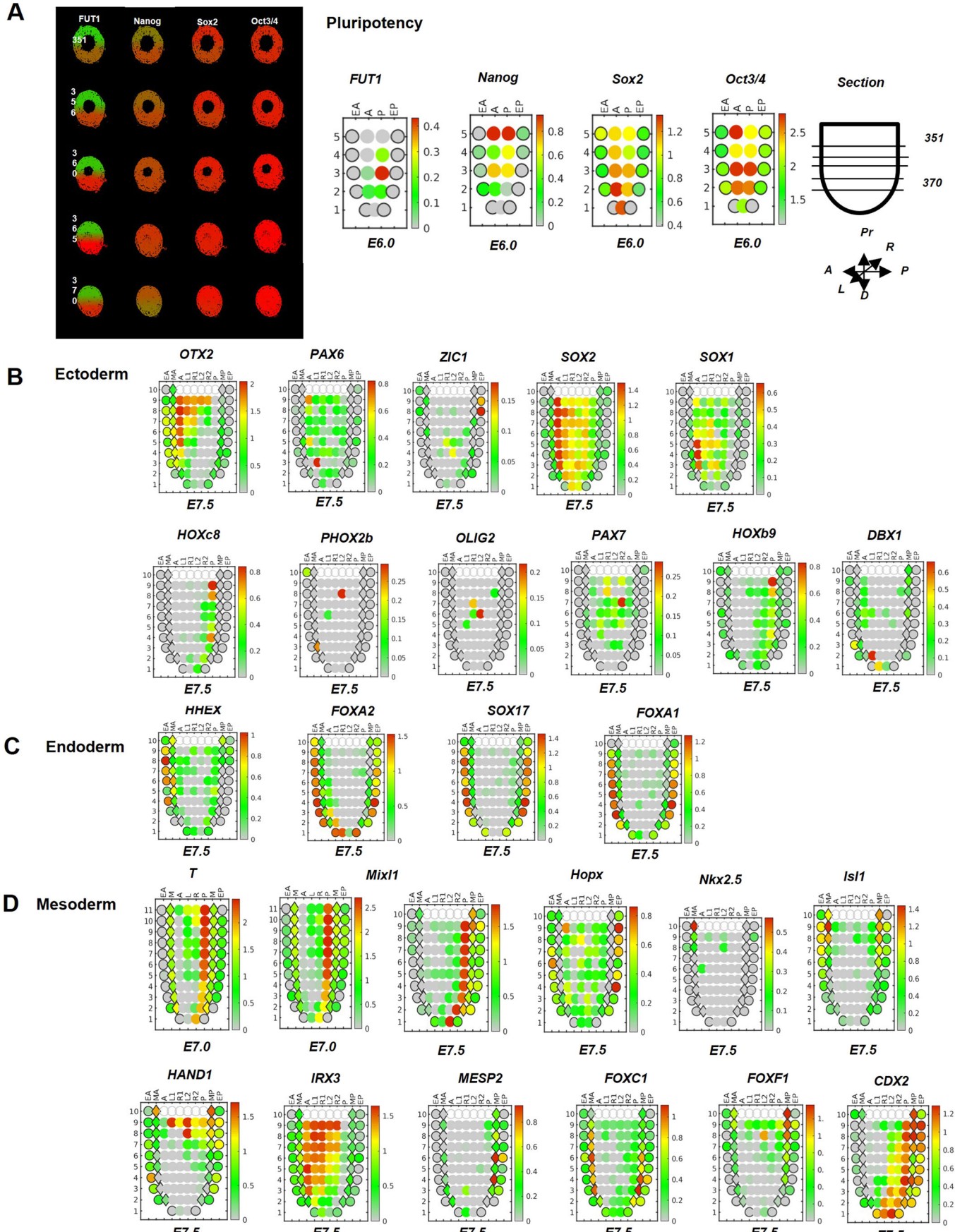

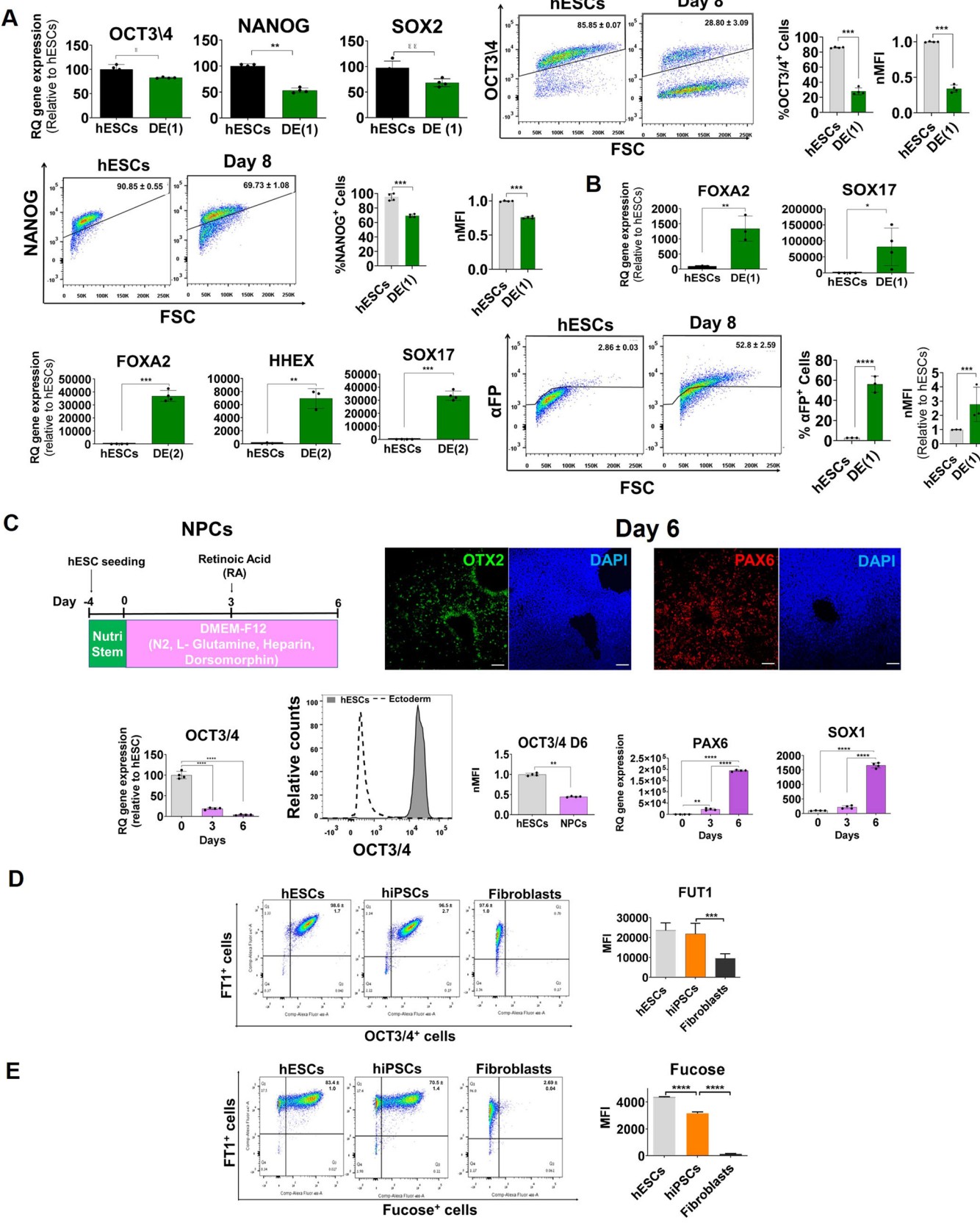

◀ **Figure EV2.  FUT1 is downregulated during PC differentiation into the tri-germ layer lineages.**

(A) Up at left: Quantification of pluripotent gene expression as measured by qPCR. Up at right: Quantification of the percentage of hESCs and hESCs-derived DE on day 8 showing the expression of OCT3/4 in histograms, bar chart, and MFI relative to hESCs. Bottom: Quantification of the percentage of hESCs and hESCs-derived DE on day 8 showing the expression of NANOG in histograms, bar chart, and MFI relative to hESCs. $n = 4$ technical replicates. (B) Up and bottom at left: Quantification of DE-specific markers, FOXA2 and SOX17 as measured by qPCR after hESC differentiation into DE using protocol 1, and FOXA2, SOX17, and HHEX as measured by qPCR after hESC differentiation into DE using protocol 2 ($n = 3$ technical replicates). Bottom at right: Quantification of the percentage of hESCs and hESCs-derived DE on day 8 showing the expression of α-FP in histograms and bar chart and MFI relative to hESCs. (C) Up at left: Schematic illustration of hESC differentiation into NPCs via bFGF and RA signaling and BMP inhibition over 6 d. Up at right: Representative immunofluorescent cultures illustrate day 6 NPCs expressing OTX2 (green) and PAX6 (red). Nuclei are stained with DAPI (blue). Scale bars represent 100 μm. Bottom: qRT-PCR and flow cytometry analyses of OCT3/4, PAX6, SOX1. The expression of FUT1 and α-fucose in iPSCs during pluripotency and differentiation is identical to that of hESCs (D) Quantification of the percent of positive cells expressing both the FT1 and OCT3/4 proteins and MFI in hESCs, a pool of hiPSCs and a pool of human fibroblasts. Pooled sample, $n = 3$ cell lines. $n = 3$ technical replicates. (E) Quantification of the percent of positive cells expressing FT1 protein and α-fucose residues and MFI in hESCs, a pool of hiPSCs, and a pool of human fibroblasts. Pooled sample, $n = 3$ cell lines. $n = 3$ technical replicates. Data information: In qPCR (A, B), data are presented as means ± SD. Two-tailed Student's t-tests *$p < 0.05$, **$p < 0.01$, ***$p < 0.001$, ****$p < 0.0001$. In FACS (A–E), data are presented as means ± SD. Ordinary One-way ANOVA **$p < 0.01$, ***$p < 0.001$, ****$p < 0.0001$.

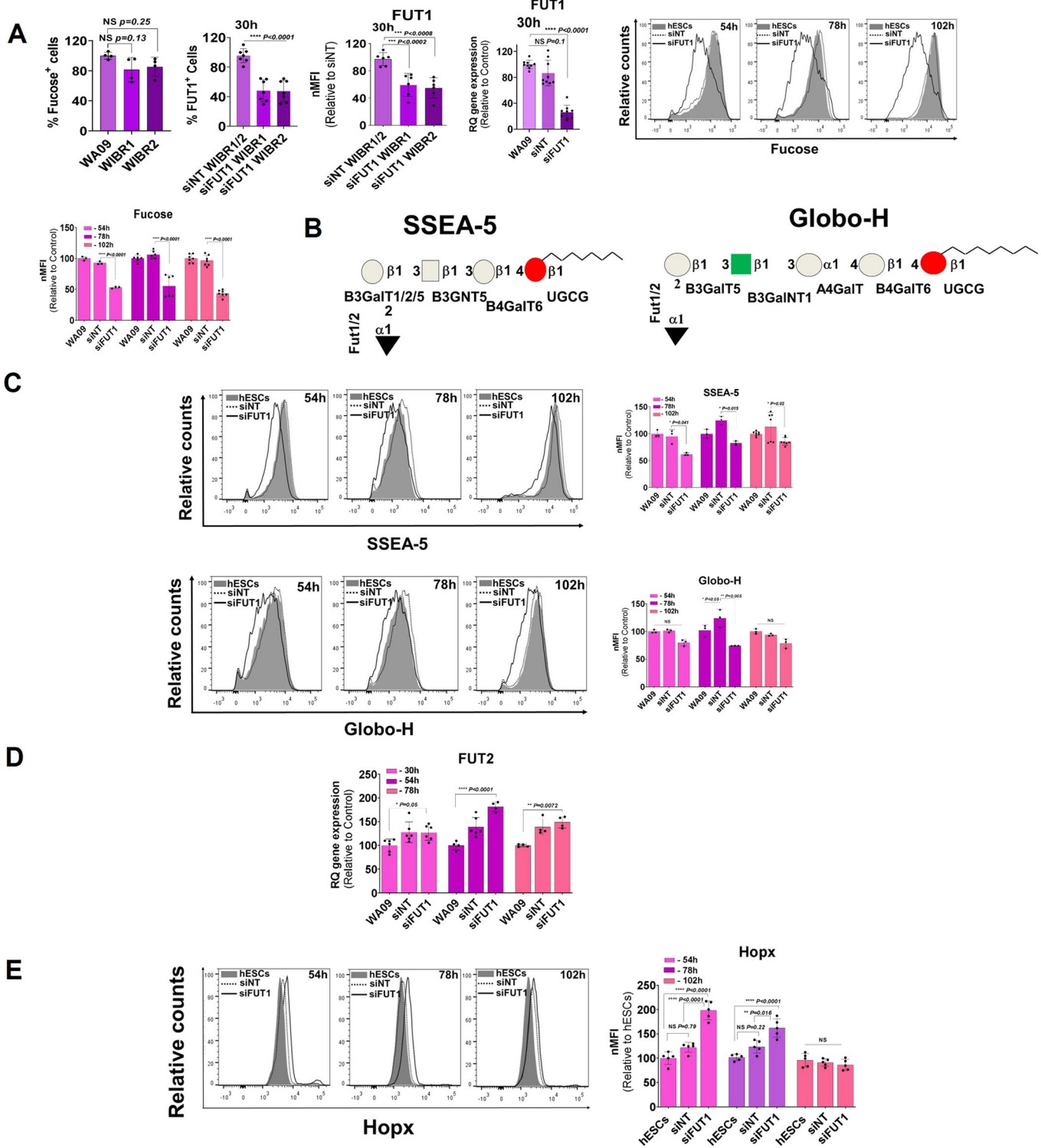

**Figure EV3.   Silencing of FUT1 alters mesoderm gene expression.**

(A) Top at left: Quantification of the percent of positive cells expressing α-fucose residues during pluripotency in three lines of hESCs: WA09, WIBR1, and WIBR2, as measured in FACS. Top at center: Quantification of the percent of positive cells expressing FT1 protein and MFI of WIBR1 and WIBR2 hESCs 30 h after FUT1 mRNA silencing (siFUT1). Negative control is WIBR1 and WIBR2 hESCs transfected with non-targeting siRNA (siNT). Top at right: Quantification of the gene expression of FUT1 in WT, siNT, and siFUT1, WA09 hESCs 30 h after FUT1 mRNA silencing, as measured by qPCR, and representative histograms showing relative counts of WT, siNT, and siFUT1, WA09 hESCs expressing α-fucose residues 54, 78, and 102 h after FUT1 mRNA silencing. Bottom: MFI of WT, siNT, and siFUT1, WA09 hESCs expressing α-fucose residues 54, 78, and 102 h after FUT1 mRNA silencing. (B) Schematic of Globo-H and SSEA-5 structures encompassing the α1,2-fucose end residue with corresponding GTs. (C) Top: Representative histograms showing relative counts of WT, siNT, and siFUT1, WA09 hESCs expressing SSEA-5 and MFI 54, 78, and 102 h after FUT1 mRNA silencing. Bottom: Representative histograms showing relative counts of WT, siNT, and siFUT1, WA09 hESCs expressing Globo-H and MFI 54, 78, and 102 h after FUT1 mRNA silencing. (D) Quantification of the gene expression of FUT2 in WT, siNT, and siFUT1, WA09 hESCs 30, 54, and 78 h after FUT1 mRNA silencing, as measured by qPCR. (E) Representative histograms showing relative counts of WT, siNT, and siFUT1, WA09 hESCs expressing HOPX and MFI 54, 78, and 102 h after silencing. $n = 4$ technical replicates. Data information: In (A, C, E), data are presented as means ± SD. Ordinary One-way ANOVA *$p < 0.05$, **$p < 0.01$, ***$p < 0.001$, ****$p < 0.0001$ or non-significant (NS). In (D), data are presented as means ± SEM. Two-tailed Student's t-tests *$p < 0.05$, **$p < 0.01$, ****$p < 0.0001$.

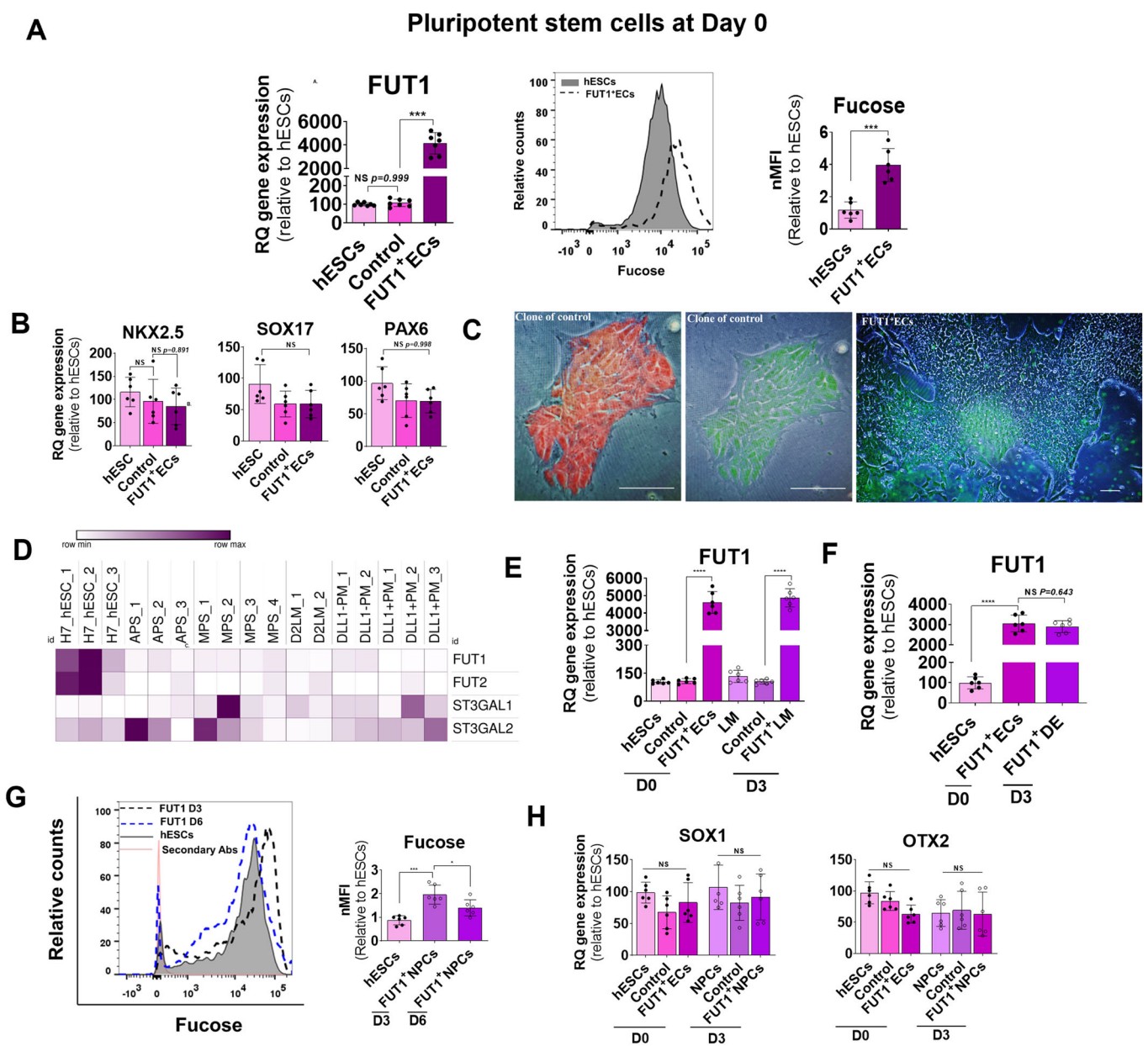

**Figure EV4. Continuous expression of FUT1 amplifies the α-fucosyl glycoconjugates and impairs hESC differentiation.**

(**A**) Left: Quantification of FUT1 transcripts in WT, mock-transfected (control ECs), and FUT1-transfected (FUT1⁺ECs), WA09 hESCs in 3 clones for each line after transfection. Right: Representative histograms showing relative counts and MFI of WT hESCs and FUT1⁺ECs expressing α-fucose. $n = 3$ technical replicates for each line. (**B**) Quantification of representative genes of the tri-germ layers, NKX2.5, SOX17, and PAX6, in WT hESCs and in 3 clones of control ECs and FUT1⁺ECs for each line showing unchanged genes after transfection. (**C**) Left and center: Representative brightfield and endogenous fluorescence protein (red, left) and (green, center) images showing clone of control ECs 18 d after culturing sorted cells. Right: Representative bright-field and endogenous fluorescence protein (green) image showing clone of FUT1⁺ECs 18 d after culturing sorted cells. Scale bars represent 100 μm. $n = 3$ technical replicates. (**D**) Heatmap of subset of RNA-seq-based gene expression profiles showing low expression of FUT1 1 d after H7 hESC differentiation into tissues that differentially expressed GT genes. Anterior primitive streak (APS), day 1 mid PS (MPS), and day 2 LM, PM, DLL⁻, and DLL⁺, PM cells. $n = 3$ per group. (**E**) Quantification of the mRNA expression levels of FUT1 in WT hESCs, control ECs, and FUT1⁺ECs on day 0 and after differentiation into LM for 3 d showing high and constant expression of FUT1. (**F**) Quantification of the mRNA expression levels of FUT1 in WT hESCs and FUT1⁺ECs on day 0 and after differentiation into DE for 3 d showing high and constant expression of FUT1. $n = 3$ technical replicates. Error bars represent ± SDs. (**G**) Representative histograms of relative counts and MFI of WT hESCs and FUT1⁺ECs showing the expression of α-fucose residues in WT hESCs and FUT1⁺ECs after differentiation into NPCs for 3 d and 6 d. (**H**) Quantification of NE markers showing SOX1 and OTX2 mRNA expression in WT hESCs, control ECs and FUT1⁺ECs on day 0 and after differentiation into NPCs for 3 d. In all experiments, $n = 3$ technical replicates. Data information: In (**A**, **G**), data are presented as means ± SD. Ordinary One-way ANOVA **$P < 0.01$, ***$P < 0.001$. In (**A**, **B**, **E**, **H**), data are presented as means ± SEM. Two-tailed Student's t-tests ****$p < 0.0001$ or non-significance (NS).

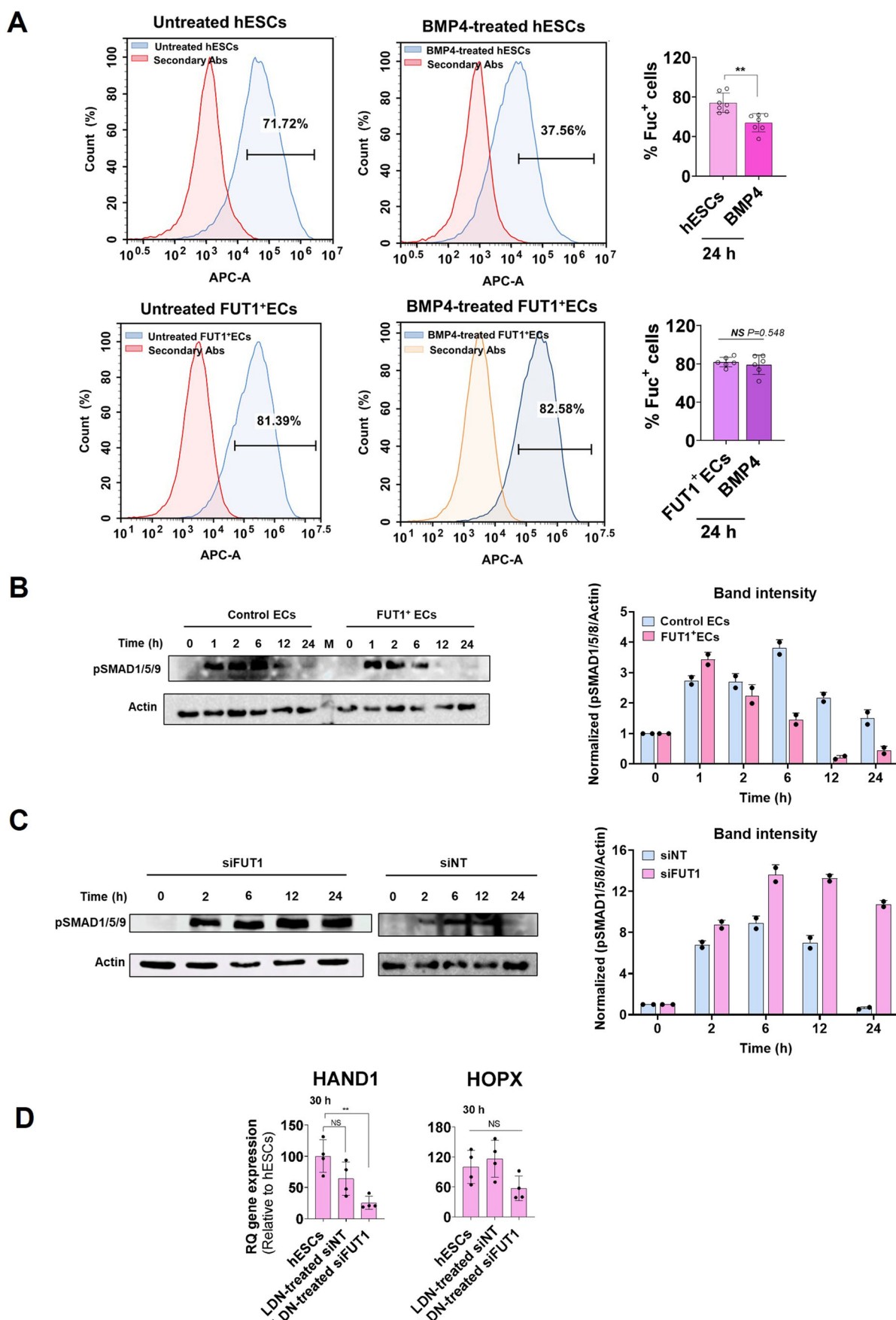

**Figure EV5.  Constitutive expression of α-fucosyl end groups interferes with BMP signaling.**

(A) Top left: Representative histograms showing relative counts of WT hESCs expressing α-fucose residues before and after BMP4 treatment for 24 h. Top right: Quantification of the percent of positive WT hESCs expressing α-fucose residues before and after BMP4 treatment for 24 h. $n = 3$ clones for each clone $n = 2$ technical replicates, Bottom left: Representative histograms showing relative counts of FUT1+ECs expressing α-fucose residues before and after BMP4 treatment for 24 h. Bottom right: Quantification of the percent of positive FUT1+ECs expressing α-fucose residues before and after BMP4 treatment for 24 h. $n = 3$ clones for each clone $n = 2$ technical replicates. (B) Left: Western blots of pSMAD1/5/8 and β-actin showing a time-course of Smad1/5/8 activity within FUT1+ECs and control ECs after BMP4 stimulation for 24 h. Right: Quantification of the pSMAD1/5/8 intensities. The normalized values are relative to normalized β-actin. $n = 2$ technical replicates. (C) Left: Western blots of pSMAD1/5/8 and β-actin showing a time-course of Smad1/5/8 activity within silenced FUT1 and siNT cells after BMP4 stimulation for 24 h. Right: Quantification of the pSMAD1/5/8 intensities. The normalized values are relative to normalized β-actin. $n = 2$ technical replicates. (D) Quantification of HAND1 and HOPX 30 h after FUT1 silencing and 6 h after LDN193189 stimulation in WT, siNT, and siFUT1 hESCs relative to WT hESCs. $n = 4$ biological replicates. Data information: In (A, D), data are presented as means ± SD. Two-tailed Student's t-tests **$p < 0.01$ or non-significance (NS).

