## [Peer Review File · EMBO Reports]

Terminal α 1,2-fucosylation of glycosphingolipids by FUT1 is a key regulator in early cell-fate decisions

Rachel Lichtenstein, Saray Chen, Dana Hayoun-Neeman, Michal Nagar, Sapir Pinyan, Limor Hadad, Liat Yaacobov, Lilach Alon, Liraz Shachar, Tair Swissa, Olga Kryukov, Orly Gershoni-Yahalom, Benyamin Rosental, and Smadar Cohen

Corresponding author(s): Rachel Lichtenstein (ruha@bgu.ac.il)

Review Timeline:

Submission Date:	13th Nov 23
Editorial Decision:	20th Dec 23
Revision Received:	29th Apr 24
Editorial Decision:	21st Jun 24
Revision Received:	9th Aug 24
Accepted:	23rd Aug 24

Editor: Achim Breiling

Transaction Report:

Dear Dr. Lichtenstein,

Thank you for the transfer of your research manuscript to EMBO reports. I have now received the reports from two of the three referees that were asked to evaluate your study, which can be found at the end of this email. A third referee had accepted to evaluate the study, but did not submit a report so far, despite several reminders. In the interest of time, I decided to proceed with the submission.

As you will see, both referees think that the findings are of interest. However, they have several comments, concerns, and suggestions, indicating that a major revision of the manuscript is necessary to allow publication of the study in EMBO reports. As the reports are below, and all the referee concerns need to be addressed, I will not detail them here, but, as is apparent from the reports, a better validation of the identified mRNAs and interactions seems necessary to consider the manuscript for publication in EMBO reports.

Given the constructive referee comments, I would like to invite you to revise your manuscript with the understanding that all referee concerns must be addressed in the revised manuscript or in a detailed point-by-point response. Acceptance of your manuscript will depend on a positive outcome of a second round of review. It is EMBO reports policy to allow a single round of revision only and acceptance of the manuscript will therefore depend on the completeness of your responses included in the next, final version of the manuscript.

1) a .docx formatted version of the final manuscript text (including legends for main figures, EV figures and tables), but without the figures included. Figure legends should be compiled at the end of the manuscript text.

2) individual production quality figure files as .eps, .tif, .jpg (one file per figure), of main figures (up to 8) and EV figures. Please upload these as separate, individual files upon re-submission.

4) a complete author checklist, which you can download from our author guidelines

(<https://www.embopress.org/page/journal/14693178/authorguide>). Please insert page numbers in the checklist to indicate where the requested information can be found in the manuscript. The completed author checklist will also be part of the RPF.

Please also follow our guidelines for the use of living organisms, and the respective reporting guidelines:
<http://www.embopress.org/page/journal/14693178/authorguide#livingorganisms>

5) that primary datasets produced in this study (e.g. RNA-seq, ChIP-seq, structural and array data) are deposited in an appropriate public database. If no primary datasets have been deposited, please also state this in a dedicated section (e.g. 'No primary datasets have been generated and deposited'), see below.

The accession numbers and database should be listed in a formal "Data Availability" section (placed after Materials & Methods) that follows the model below. This is now mandatory (like the COI statement). Please note that the Data Availability Section is restricted to new primary data that are part of this study. This section is mandatory. As indicated above, if no primary datasets have been deposited, please state this in this section

Data availability

8) Regarding data quantification and statistics, please make sure that the number "n" for how many independent experiments were performed, their nature (biological versus technical replicates), the bars and error bars (e.g. SEM, SD) and the test used to calculate p-values is indicated in the respective figure legends (also for potential EV figures and all those in the final Appendix). Please also check that all the p-values are explained in the legend, and that these fit to those shown in the figure. Please provide statistical testing where applicable. Please avoid the phrase 'independent experiment', but clearly state if these were biological or technical replicates. Please also indicate (e.g. with n.s.) if testing was performed, but the differences are not significant. In case n=2, please show the data as separate datapoints without error bars and statistics. See also: <http://www.embopress.org/page/journal/14693178/authorguide#statisticalanalysis>

9) Please add scale bars of similar style and thickness to all the microscopic images, using clearly visible black or white bars (depending on the background). Please place these in the lower right corner of the images themselves. Please do not write on or near the bars in the image but define the size in the respective figure legend.

10) Please also note our reference format:
<http://www.embopress.org/page/journal/14693178/authorguide#referencesformat>

12) We now use CRediT to specify the contributions of each author in the journal submission system. CRediT replaces the author contribution section. Please use the free text box to provide more detailed descriptions and do not provide your final

manuscript text file with an author contributions section. See also our guide to authors:
<https://www.embopress.org/page/journal/14693178/authorguide#authorshipguidelines>

13) We would encourage you to use 'Structured Methods', our new Materials and Methods format. According to this format, the Materials and Methods section should include a Reagents and Tools Table (listing key reagents, experimental models, software and relevant equipment and including their sources and relevant identifiers), uploaded as separate file, followed by a Methods and Protocols section in which we encourage the authors to describe their methods using a step-by-step protocol format with bullet points, to facilitate the adoption of the methodologies across labs. More information on how to adhere to this format as well as downloadable templates (.doc or .xls) for the Reagents and Tools Table can be found in our author guidelines (section 'Structured Methods'):

14) Please add up to 5 keywords to the manuscript and order the sections like this, using these names:
Title page - Abstract - Keywords - Introduction - Results - Discussion - Materials and Methods - Data availability section - Acknowledgements - Disclosure and Competing Interests Statement - References - Figure legends - Expanded View Figure legends

Please note that all corresponding authors are required to supply an ORCID ID for their name upon submission of a revised manuscript. Please find instructions on how to link the ORCID ID to the account in our manuscript tracking system in our Author guidelines: <http://www.embopress.org/page/journal/14693178/authorguide#authorshipguidelines>

I look forward to seeing a revised version of your manuscript when it is ready. Please let me know if you have questions or comments regarding the revision.

Yours sincerely,

Referee #1:

This report focuses on changes in GSL expression that may serve as mediators of early embryonic developmental events. In large part, the authors have met this goal and have identified a key reciprocal inter-relationship between expression of a(1,2)-fucosylated GSLs and BMP4 signaling, whereby increased cell surface expression of a(1,2)-fucosylated structures dampens BMP-4-mediated signaling and BMP-4-mediated signaling dampens expression of a(1,2)-fucosylated GSLs. This molecular toggle has implications for the differentiation of ESCs into mesodermal-lineage cells, in particular, myocardial cells.

The challenge inherent to understanding the results of this report is that the Introduction does not adequately frame the intended study. In particular, the complexity of the structural biology of glycoconjugates merits that a significant amount of background information be presented in an Introduction section of a report on such a subject. As such, in my view, it is very challenging for someone who is not versed in glycoscience to comprehend both the implications of this report and its limitations.

While I am not suggesting how to rewrite the Introduction, I believe that the Introduction could benefit from a formally stated hypothesis. Instead, the last sentence of this report poses a question: "Does one sugar in the terminal position of a lipid glyco-series enable the alteration of genetic reprogramming?" It seems that the authors purposely avoided enunciating a hypothesis in order to frame that the investigative approach is "unbiased". However, to address this question with the most "unbiased" approach, one must perform MS analysis of cell surface glycans to fully elucidate the breadth and scope of the target monosaccharide substitution. But, this task is not possible if one does not have copious amounts of cell numbers (i.e., starting material), and the amount of available biologic material is extremely limited in the study of cells within mouse embryos at different stages of development. Instead, then, one is left with use of lectins and antibodies as probes. Once these reagents are employed, the study is not "unbiased" because these probes each introduce biases. Given all this, I offer the following considerations:

MAJOR COMMENTS:

(1) On page 5, it is written: "We then searched for a model that could confirm the STAINING results to obtain ADDITIONAL UNBIASED EVIDENCE..." (caps added by the Reviewer). Indeed, if one wishes to "confirm" the findings of "unbiased evidence", then, by intention, a "confirmatory" action is "biased" because one should be "assessing" -- not "confirming" -- since the intended subsequent exploratory action is supposed to be "unbiased".

(2) In Figure 1A, I suggest not placing the sialic acids within that diagram -- confine the depicted structures to only SSEA-5 and Globo-H (e.g., if you wish to depict SSEA-4 to contrast it to Globo-H, then draw the SSEA-4 structure separately).

(3) Nowhere in the report is there mention of the limitations of UEA-1 lectin to detect "alpha-fucose" (a-fucose). It is well known that UEA-1 lectin almost exclusively detects $\alpha(1,2)$ -fucosylated Type 2 lactosamines (and, importantly, especially detects the structure Lewis Y (LeY)), but does not detect $\alpha(1,2)$ -fucosylated Type 1 lactosamines (such as SSEA-5) nor any $\alpha(1,2)$ -fucosylated globo-series structures (such as Globo-H). Thus, Figure 4B does not provide adequate information, as the impact of siRNA to FUT1 on expression of $\alpha(1,2)$ -fucosylated GSLs should be measured in context of a staining shift between Globo-H against SSEA-3. Indeed, I strongly suggest that FACS histograms be shown for the staining level shifts be shown for UEA-1, Globo-H, SSEA-3, and SSEA-5 for all data regarding knock-down of FUT1 (Figure 4B) or overexpression of FUT1 (i.e., Figure 5A).

(4) The authors are commended for specifically measuring FUT2 and SEC in the mouse models. The authors must mention that SEC is a pseudogene in the human, so this $\alpha(1,2)$ -FT is not a consideration in human cells. However, the authors must reconcile their data with published data by J Chen et al (DOI: <https://doi.org/10.1101/615070>) indicating that triple knock-out of these 3 genes in mice leads to no developmental abnormalities. Ideally, future studies by these authors could employ triple knock-down or CRISPR editing of these genes to provide a greater understanding of the role of $\alpha(1,2)$ -fucosylated structures in embryonic development.

MINOR COMMENTS:

(1) The gene name should always be in italics (i.e., italicize FUT if you are referring to the gene or the transcript). In contrast, it is best to not call fucosyltransferase proteins by "FUT" as "FUT" can be confused with the gene name. Instead, I recommend using "FT" to indicate the protein, i.e., "FT1" and "FT2".

(2) The use of embryonic cells from mouse or human must be clarified at every instance when reference to an embryonic stem cell is made. For example, at the outset of the Results, it should be made clear that studies are being performed on "isolated MOUSE embryos" (not "isolate embryos").

(3) On page 6, the sentence starting with "In line with..." belongs in the Discussion, with a greater development of this notion and with responsiveness to the finding of Chen (see above).

(4) The methods to study human ESCs employ human ESC lines. This fact should be emphasized throughout the Results. Better yet, emphasize within Results the human ES cell line that was used for each analysis.

(5) On page 9, the sentence "The hiPSCs expressed SSEA-4 and other stem cell genes..." should read: "The hiPSCs expressed the GSL SSEA-4 and genes associated with stem cells in a manner slightly differently than that observed in hESC lines." Clearly, the authors know that SSEA-4 is not a gene, but this sentence is misleading.

(6) On page 15, the phrase that states "ABO blood group antigens" should read as follows: "ABO blood group antigens, Lewis antigens, Globo-H and SSEA-5".

Referee #2:

In the manuscript Chen et al. report a set of glycosphingolipids (GSLs) that may regulate early cell-fate decisions. The authors employ hESC lines where Fut1, which catalyzes addition of α -linked fucosyl residues to GSLs, can be over-expressed or knocked-down to investigate its effects on pluripotency state and differentiation into various progenitor fates. The authors show that in hESCs, down-regulation of Fut1 (and, consequently, $\alpha(1,2)$ -fucosyl residues) causes a transient decrease in expression of pluripotency factors Oct4 and Nanog. In turn, up-regulation of Fut1 in hESCs differentiating towards mesoderm reduces expression of LM genes, many of which are targets of Bmp4 signaling pathway. The authors conclude that high levels of $\alpha(1,2)$ -fucosyl glycoconjugates are required to maintain the pluripotency state and impair hESC commitment to progenitor fates. In particular, the authors claim that $\alpha(1,2)$ -fucosyl GSLs inhibit BMP signaling, thereby disrupting mesodermal differentiation.

The experiments presented in this study are thorough and well annotated. Statistical tests are, for the most part, performed to demonstrate significant changes in expression levels. The gained insights are novel and relevant for the field.

However, several main conclusions drawn by the authors (including the suggestion that α -fucosyl groups are targets and regulators of BMP pathway, and the suggested roles α -fucosyl groups may play in regulating pluripotency state and early cell-fate decisions) are in my opinion based on correlative analyses that do not imply causation and therefore are not entirely supported by the currently presented evidence. I suggest some clarifications and additional experiments that could potentially improve the quality of the manuscript.

Major comments:

1. In several sections of the manuscript, the authors attribute the effects of Fut1 over-expression/knock-down to the consequent up-/down-regulation of alpha1,2-fucosyl GSLs. Can the authors demonstrate that the effects of altering the Fut1 levels are mediated directly and exclusively by alpha1,2-fucosyl GSLs and can they exclude that in hESCs and in differentiated cells Fut1 may fucosylate substrates other than GSLs?

Reduction or increase of alpha1,2-fucosyl GSLs is one of the effects of knocking-down or over-expressing Fut1, that occurs simultaneously or even after the events that authors describe in the manuscript. For example, in the section "alpha 1,2-fucosyl GSLs are hallmarks of pluripotency" the authors silence Fut1 in three lines of hESCs and observe reduction of Oct3/4 and Nanog expression levels at 30h (Fig. 4E). However, low expression of alpha-fucosyl residues is first detected at 54h (Fig. S4A). I suggest the authors to include the analysis of alpha-fucosyl residues at earlier time points (e.g., 30h or earlier for this example).

2. The authors claim that high levels of alpha1,2-fucosyl GSLs (resulting from over-expression of Fut1) are "required to maintain the pluripotency state", and that "large amount of FUT1 transcripts and protein are needed to maintain cell pluripotency" and FUT1 is a "crucial enzyme required in the pluripotency state". At the same time, the authors report that upon down-regulation/silencing of Fut1, hESCs maintained undifferentiated morphology and "did not lose pluripotency in the long term". The existence of triple knockout homozygous Fut1/Fut2/Sec1 cells (The Jackson Laboratory, B6.Cg-Fut1tm1DzhouDel(7Fut2-Sec1)1Dzhou/J; see also <https://www.biorxiv.org/content/10.1101/615070v1.full.pdf>) and their ability to generate viable and fertile mice strongly suggests that Fut1 is not required for pluripotency maintenance.

Could the authors explain the discrepancy between their claims and their own observations/public Fut1 KO ESC data?

3. The authors detect a transient minor (several orders of magnitude lower than in LM differentiation) up-regulation of Hand1, Hopx, and Smad4 upon silencing of Fut1 (Fig. 4F) and suggest that "reduced alpha1,2-fucosyl residues may facilitate BMP signals". It is an interesting hypothesis, but in my opinion the provided evidence is not sufficient to support it. I suggest the authors to perform a time-course analysis of Smad1/5/9 phosphorylation, e.g. by western blotting, upon Fut1 silencing (using siNT as control) to see if reduction of Fut1, and consequently alpha1,2-fucosyl residues, lead to increase in phospho-Smad1/5/9 levels. Secondly, the authors could use LDN193189 (which they employed elsewhere in the manuscript) to test whether Hand1 and Hopx expression levels are not affected upon Fut1 silencing in the presence of the inhibitor.

4. The authors claim that the reduction of alpha-fucosyl end groups after Bmp4 treatment (Fig. 6A) suggests that BMP4 is a "critical component in the regulation of sugar metabolism". In my opinion, the experimental evidence provided by the authors is not sufficient to prove a direct functional link between Bmp4 signaling and reduction of alpha-fucosyl end groups, but only shows that upon Bmp4-induced differentiation the levels of alpha-fucosyl end groups go down. Similar levels of alpha-fucose in hESCs and LDN193189 (Fig. 6A) is a result of LDN193189 blocking the differentiation, and do not prove direct link between Bmp4 activity and alpha-fucosyl end group levels.

alpha-fucosyl end groups, like Fut1, are reduced in all differentiation protocols tested by the authors, including neural differentiation, where Bmp4 is not active (Fig. 3J). Therefore, as the authors suggest earlier in the manuscript for Fut1, reduction of alpha-fucosyl groups appears to be a hallmark of early differentiation, not necessarily caused by Bmp4. Bmp4 induces exit from pluripotency (similarly to other cues the authors used other differentiation protocols), which in turn results in the down-regulation of Fut1 and alpha-fucosyl groups. Can Fut1 expression be controlled by pluripotency factors such as Oct4 or Nanog instead of Bmp4?

5. The authors claim that alpha1,2-fucosyl GSLs inhibit BMP4 signaling, based on the results of FACS analysis presented in Fig. 6E. The histogram displaying Fut1+ cells (Fig. 6E, middle panel) shows that even in Fut1+ cells, with high Fut1 and fucosyl group levels, a large fraction of Bmp4-treated cells are phospho-Smad1/5/8 positive. Also, this fraction appears larger than 10.57% as reported in the text. Could the quantification results shown in the graphs in Fig. 6E perhaps be affected by the arbitrary setting of the gates?

The results presented in Fig. 6 only show that cells over-expressing Fut1 have a somewhat weaker response to Bmp4 signaling compared to the control cells. This may be caused by a number of factors, for example, lower levels of BMPRs in the FUT1+ cells or variations in membrane permeability between control and Fut1+ cells. In my opinion, the claim that alpha1,2-fucosyl GSLs inhibit BMP4 signaling (title of the section) is a misinterpretation of the presented evidence.

To determine the effect of Fut1 over-expression on cellular response to Bmp4, I suggest the following experiment. The authors could quantitatively assess the levels of Smad1/5/8 phosphorylation by Western Blotting in a time-dependent manner (e.g., after 1, 2, 6, 12, 24 h of Bmp4 treatment) in control and Fut1+ cells. Analyzing these earlier time-points would also exclude any secondary effects that may have occurred within the 24 h period between Bmp4 application and cell collection in the presented experiment.

Minor comments:

1. In Fig. 4F expression of genes is assayed at different time points. For example, Hand1 at 54 and 78h. Isl1 and Nkx2.5 at 30h. Isl1 and especially Nkx2.5 are expected to be up-regulated at the same/similar time point as Hand1. Were the expression of Nkx2.5 and Isl1 assayed at 54 and 78h as well? If so, could the authors include the results in the figure? Otherwise, could the authors explain why different time point were chosen.

2. In the Discussion section the authors talk about the activation of Smad2/3 when alpha1,2-fucosyl glycans are over-expressed. This interesting hypothesis can be relatively easily checked using Western blotting, but in the experimental section I did not see

any analysis of Smad2/3 phosphorylation. I suggest the authors either perform this experiment or exclude the speculation of Smad2/3 activation from the Discussion.

3. Fig. 4E. Only Nanog expression is shown at 30h and 54h. Oct4 and Sox2 expressions are only shown at 30h. What does Oct4 and Sox2 expression at 54h look like?

4. Fig. 4F. For all genes except for Smad4 the p values are calculated for siNT vs siFUT1. Is the difference between Smad4 expression in siNT and siFUT1 statistically significant? Similarly, Fig. S4E should show p values for siNT vs siFUT1.

5. There are some inconsistencies in terminology throughout the manuscript. For example: α 1,2-fucosyl vs α 1,2-fucosyl (extra space); globoseries vs globo-series; Oct3\4 (Fig. 4E) vs Oct3/4 (Fig. 3I), etc.

EMBOR-2023-58438-T

Dear editor and reviewers,

We'd like to thank the editor for the opportunity to submit our amended paper, as well as the reviewers for their feedback and recommendations that helped improve it. We handled the reviewer's helpful comments conceptually and empirically. A complete, point-by-point response to the letter is included below. Sentences and paragraphs added to correct faults in the amended version are underlined yellow. We trust that the new work will suit your journal's requirements.

Detailed point-by-point response to reviewer's comments**Referee #1:**

The challenge inherent to understanding the results of this report is that the Introduction does not adequately frame the intended study. In particular, the complexity of the structural biology of glycoconjugates merits that a significant amount of background information be presented in an Introduction section of a report on such a subject. As such, in my view, it is very challenging for someone who is not versed in glycoscience to comprehend both the implications of this report and its limitations.

While I am not suggesting how to rewrite the Introduction, I believe that the Introduction could benefit from a formally stated hypothesis. Instead, the last sentence of this report poses a question: "Does one sugar in the terminal position of a lipid glyco-series enable the alteration of genetic reprogramming?" It seems that the authors purposely avoided enunciating a hypothesis in order to frame that the investigative approach is "unbiased". However, to address this question with the most "unbiased" approach, one must perform MS analysis of cell surface glycans to fully elucidate the breadth and scope of the target monosaccharide substitution. But, this task is not possible if one does not have copious amounts of cell numbers (i.e., starting material), and the amount of available biologic material is extremely limited in the study of cells within mouse embryos at different stages of development. Instead, then, one is left with use of lectins and antibodies as probes. Once these reagents are employed, the study is not "unbiased" because these probes each introduce biases. Given all this, I offer the following considerations:

Author response:

We appreciate this reviewer's constructive comment. Certainly, we wanted to be unbiased and using mass spectrometry study of cell surface glycans to completely understand the monosaccharide substitution in mouse embryos is difficult. Therefore, the final paragraph in the introduction was altered as advised, and hypothesis was added, pages 3-4.

MAJOR COMMENTS:

(1) On page 5, it is written: "We then searched for a model that could confirm the STAINING results to obtain ADDITIONAL UNBIASED EVIDENCE..." (caps added by the Reviewer). Indeed, if one wishes to "confirm" the findings of "unbiased evidence", then, by intention, a

"confirmatory" action is "biased" because one should be "assessing" -- not "confirming" -- since the intended subsequent exploratory action is supposed to be "unbiased".

Author response:

Following the reviewer's critique, we revised the sentence to – “We then searched for a model that would show the spatiotemporal expression of the FUT1 gene in embryos at an early stage of development, before emergence of the PS.”

(2) In Figure 1A, I suggest not placing the sialic acids within that diagram -- confine the depicted structures to only SSEA-5 and Globo-H (e.g., if you wish to depict SSEA-4 to contrast it to Globo-H, then draw the SSEA-4 structure separately).

Author response:

Figure 1 was altered based on the reviewer's suggestions. Since the paragraph “Changes in α 1,2-fucosyl group expression occur during embryogenesis” in page 4 describes the results for SSEA-5 and Globo-H, an illustration of the SSEA4 structure is not required. The updated Figure 1A illustrates the structures of SSEA-5 and Globo-H.

(3) Nowhere in the report is there mention of the limitations of UEA-1 lectin to detect "alpha-fucose" (a-fucose). It is well known that UEA-1 lectin almost exclusively detects a(1,2)-fucosylated Type 2 lactosamines (and, importantly, especially detects the structure Lewis Y (LeY)), but does not detect a(1,2)-fucosylated Type 1 lactosamines (such as SSEA-5) nor any a(1,2)-fucosylated globo-series structures (such as Globo-H). Thus, Figure 4B does not provide adequate information, as the impact of siRNA to FUT1 on expression of a(1,2)-fucosylated GSLs should be measured in context of a staining shift between Globo-H against SSEA-3. Indeed, I strongly suggest that FACS histograms be shown for the staining level shifts be shown for UEA-1, Globo-H, SSEA-3, and SSEA-5 for all data regarding knock-down of FUT1 (Figure 4B) or overexpression of FUT1 (i.e., Figure 5A).

Author response:

We appreciate the reviewer's feedback. Supplementary data Figures 4A and 4C depict FACS staining for UEA-1, Globo-H, and SSEA-5 after 54, 78, and 102 hours of FUT1 silencing. To be consistent with the display of transcription factor expression 30 hours after silencing, Figure 4B required FACS tagging of UEA-1, Globo-H, and SSEA-3.

As a result, the revised Figure 4B shows FACS representative histograms and MFI for UEA-1, Globo-H, and SSEA-3 30 hours after silencing. The FUT1 gene was silenced for 30 hours in WA09 cells using a pool of 3 siRNA (at concentrations of 20 nM and 50 nM) and 13.3 siRNA at 20 nM.

(4) The authors are commended for specifically measuring FUT2 and SEC in the mouse models. The authors must mention that SEC is a pseudogene in the human, so this a(1,2)-FT is not a consideration in human cells. However, the authors must reconcile their data with published data by J Chen et al (DOI: <https://doi.org/10.1101/615070>) indicating that triple knock-out of these 3 genes in mice leads to no developmental abnormalities. Ideally, future studies by these authors could employ triple knock-down or CRISPR editing of these genes to

provide a greater understanding of the role of α (1,2)-fucosylated structures in embryonic development.

Author response:

As indicated by the reviewer, we updated the wording on page 5 to clarify that SEC1 is not a factor in human cells. “Transcripts of SEC1, a GT defined **only** in mice and catalyzes α 1,2-fucosyl reaction to glycoproteins (Domino et al., 2001).”

Our results are in one line with Chen J et al. in vivo study. As well, our evidence is devoid of correlating the level of FUT1 transcripts and alpha1,2-fucosyl GSLs with the cell pluripotency. Thus, according to this reviewer’s comment, several statements in the text were amended, and Chen J et al. preprint was added to the text and bibliography. The following evidence was seen: FUT1 and alpha1,2-fucosyl GSLs decrease as hESCs are differentiated. When the enzyme and the sugars are over-expressing the ability of embryonic cells to differentiate is compromised for the short and the long term (mesoderm and CMs). When FUT1 is silenced and less alpha1,2-fucosyl GSLs are generated, many transcription factors associated with cell differentiation rise by 1.5-3-fold, while pluripotency-associated transcription factors decrease by less than 50% over a brief period of 30-54 hours. However, after 78 hours, pluripotent markers are restored, and TFs of cell differentiation decreased to undetectable levels (data not shown), excluding HOPX. Furthermore, after silencing FUT1, the hESCs retained undifferentiated morphology and did not lose pluripotency in the long term. It appears that following silencing, genetic alterations within the cells are dynamic and reversible. If more than half of the pluripotent markers were lowered, the cells would lose their pluripotency and differentiate (usually to endoderm) (Heurtier et al., 2019; Niwa et al., 2000; Pan and Thomson, 2007; Xiong et al., 2022). Therefore, the existence of triple knockout homozygous Fut1/Fut2/Sec1 mice (The Jackson Laboratory, B6.Cg Fut1tm1DzhouDel(7Fut2-Sec1)1Dzhou/J mice), and their ability to generate viable and fertile mice strengthening our in vitro observations that Fut1 is not a hallmark of pluripotency. Please see the revised text marked in yellow at pages 9, 11-12 and 16.

MINOR COMMENTS:

(1) The gene name should always be in italics (i.e., italicize FUT if you are referring to the gene or the transcript). In contrast, it is best to not call fucosyltransferase proteins by "FUT" as "FUT" can be confused with the gene name. Instead, I recommend using "FT" to indicate the protein, i.e., "FT1" and "FT2".

Author response:

We modified the protein name of fucosyltransferase into FT.

(2) The use of embryonic cells from mouse or human must be clarified at every instance when reference to an embryonic stem cell is made. For example, at the outset of the Results, it should be made clear that studies are being performed on "isolated MOUSE embryos" (not "isolate embryos").

Author response:

We agree with the reviewer, using only embryo may confuse the reader. We indicated in the text that we isolate and study mouse embryos.

(3) On page 6, the sentence starting with "In line with..." belongs in the Discussion, with a greater development of this notion and with responsiveness to the finding of Chen (see above).

Author response:

"In line with..." was revised as the reviewer suggested.

(4) The methods to study human ESCs employ human ESC lines. This fact should be emphasized throughout the Results. Better yet, emphasize within Results the human ES cell line that was used for each analysis.

Author response:

In the result part, we used the term hESCs, but the exact lines that were used are indicated in the figure legends.

(5) On page 9, the sentence "The hiPSCs expressed SSEA-4 and other stem cell genes..." should read: "The hiPSCs expressed the GSL SSEA-4 and genes associated with stem cells in a manner slightly differently than that observed in hESC lines." Clearly, the authors know that SSEA-4 is not a gene, but this sentence is misleading.

Author response:

The sentence was revised as the reviewer suggested.

(6) On page 15, the phrase that states "ABO blood group antigens" should read as follows: "ABO blood group antigens, Lewis antigens, Globo-H and SSEA-5".

Author response:

The sentence was revised as the reviewer suggested.

Referee #2:

In the manuscript Chen et al. report a set of glycosphingolipids (GSLs) that may regulate early cell-fate decisions. The authors employ hESC lines where Fut1, which catalyzes addition of α -linked fucosyl residues to GSLs, can be over-expressed or knocked-down to investigate its effects on pluripotency state and differentiation into various progenitor fates. The authors show that in hESCs, down-regulation of Fut1 (and, consequently, α 1,2-fucosyl

residues) causes a transient decrease in expression of pluripotency factors Oct4 and Nanog. In turn, up-regulation of Fut1 in hESCs differentiating towards mesoderm reduces expression of LM genes, many of which are targets of Bmp4 signaling pathway. The authors conclude that high levels of α 1,2-fucosyl glycoconjugates are required to maintain the pluripotency state and impair hESC commitment to progenitor fates. In particular, the authors claim that α 1,2-fucosyl GSLs inhibit BMP signaling, thereby disrupting mesodermal differentiation.

The experiments presented in this study are thorough and well annotated. Statistical tests are, for the most part, performed to demonstrate significant changes in expression levels. The gained insights are novel and relevant for the field.

However, several main conclusions drawn by the authors (including the suggestion that α -fucosyl groups are targets and regulators of BMP pathway, and the suggested roles α -fucosyl groups may play in regulating pluripotency state and early cell-fate decisions) are in my opinion based on correlative analyses that do not imply causation and therefore are not entirely supported by the currently presented evidence. I suggest some clarifications and additional experiments that could potentially improve the quality of the manuscript.

Major comments:

1. In several sections of the manuscript, the authors attribute the effects of Fut1 over-expression/knock-down to the consequent up-/down-regulation of alpha1,2-fucosyl GSLs. Can the authors demonstrate that the effects of altering the Fut1 levels are mediated directly and exclusively by alpha1,2-fucosyl GSLs and can they exclude that in hESCs and in differentiated cells Fut1 may fucosylate substrates other than GSLs?

Author response –

The reviewer raised an important point. FUT1 is glycosyltransferase that is involved in the creation of a precursor of the H antigen, which is required for the final step in the synthesis of soluble A and B antigens. It may add fucose residue in α 1,2 linkage to end structures of GSLs as well as to glycoproteins. Based on findings published by Kawamura T. et al., (2014, PLOS One, 9(10): e111064 and 2015, Stem Cell Translational Medicine, 4: 1258-1264), we understood that hESCs, hiPSCs and immature cardiomyocytes (CMs) do not synthesize α 1,2-fucosyl glycan structures in glycoproteins. Following the reviewer's comment, we checked the literature for additional publications regarding α 1,2-fucosyl synthesis and found that these specific fucosyl groups were observed in glycoproteins of mature heart cells (Ashwood C., 2020, J Mol Cell Cardiol., 139: 33-46) but not in glycoproteins of pluripotent cells and not in earlier stages of iPSC-derived CMs, cells which are relevant to our study. Therefore, our flow cytometry results of the structures, SSEA-3, SSEA5 and Globo-H present the Globo series and Lacto series of GSLs and FUT1 in our study is involved in the synthesis of GSLs. We addressed this remark at the end of the introduction section, on page 4, and included references that discussed it.

Reduction or increase of alpha1,2-fucosyl GSLs is one of the effects of knocking-down or

over-expressing Fut1, that occurs simultaneously or even after the events that authors describe in the manuscript. For example, in the section " α 1,2-fucosyl GSLs are hallmarks of pluripotency" the authors silence Fut1 in three lines of hESCs and observe reduction of Oct3/4 and Nanog expression levels at 30h (Fig. 4E). However, low expression of α -fucosyl residues is first detected at 54h (Fig. S4A). I suggest the authors to include the analysis of α -fucosyl residues at earlier time points (e.g., 30h or earlier for this example).

Author response –

As advised by the reviewer, FUT1 in WA09 was silenced for 30 hours with a pool of 3 siRNA (at two doses of 20 nM and 50 nM) and 13.3 siRNA (20 nM). Figure 4A shows FACS typical histograms and MFI for α -fucosyl residues (UEA-I) 30 hours after FUT1 mRNA silencing.

2. The authors claim that high levels of alpha1,2-fucosyl GSLs (resulting from over-expression of Fut1) are "required to maintain the pluripotency state", and that "large amount of FUT1 transcripts and protein are needed to maintain cell pluripotency" and FUT1 is a "crucial enzyme required in the pluripotency state". At the same time, the authors report that upon down-regulation/silencing of Fut1, hESCs maintained undifferentiated morphology and "did not lose pluripotency in the long term".

The existence of triple knockout homozygous Fut1/Fut2/Sec1 cells (The Jackson Laboratory, B6.Cg-Fut1tm1DzhouDel(7Fut2-Sec1)1Dzhou/J; see also <https://www.biorxiv.org/content/10.1101/615070v1.full.pdf>) and their ability to generate viable and fertile mice strongly suggests that Fut1 is not required for pluripotency maintenance.

Could the authors explain the discrepancy between their claims and their own observations/public Fut1 KO ESC data?

Author response –

We appreciate the reviewer's critique. We agree that our evidence is devoid of correlating the level of FUT1 transcripts and alpha1,2-fucosyl GSLs with the cell pluripotency. The following evidence was seen: FUT1 and alpha1,2-fucosyl GSLs decrease as hESCs are differentiated. When the enzyme and the sugars are over-expressing the ability of embryonic cells to differentiate is compromised for the short and the long term (mesoderm and CMs). When FUT1 is silenced and less alpha1,2-fucosyl GSLs are generated, many transcription factors associated with cell differentiation rise by 1.5-3-fold, while pluripotency-associated transcription factors decrease by less than 50% over a brief period of 30-54 hours. However, after 78 hours, pluripotent markers are restored, but TFs of cell differentiation decreased to undetectable levels. Furthermore, after silencing FUT1, the hESCs retained undifferentiated morphology and did not lose pluripotency in the long term. It appears that following silencing, genetic alterations within the cells are dynamic and reversible. If more than half of the pluripotent markers were lowered, the cells would lose their pluripotency and differentiate (usually to endoderm) (Heurtier et al., 2019; Niwa et al., 2000; Pan and Thomson, 2007; Xiong et al., 2022). Furthermore, we read a new opinion paper by Andrews and Gokhale, 2024, which claims that pluripotent markers such as Globo-H and SSEA-1-5 are not specific to pluripotent stem cells, but are also expressed by somatic cells, and thus are not a hallmark

of pluripotency. As well, the existence of triple knockout homozygous Fut1/Fut2/Sec1 mice (The Jackson Laboratory, B6.Cg Fut1tm1DzhouDel(7Fut2-Sec1)1Dzhou/J mice), and their ability to generate viable and fertile mice strengthening our *in vitro* observations that Fut1 is not a hallmark of pluripotency. We assume that after Fut1/Fut2/Sec1 KO, critical transcription factors such as OCT3/4 transiently diminished inside blastocytes after fertilization, then recovered, enabling for the generation of viable and fertile mice based on FUT1/FUT2/SEC1 knockout. According to our findings in this study, if Chen J et al. overexpressed FUT1 during blastocyst micromanipulation, we believe they would have difficulty producing mouse embryos.

According to this reviewer's comment, several statements in the text were amended, and the mentioned preprint was added to the text and bibliography. Please see the revised text marked in yellow at pages 9, 11-12 and 16.

3. The authors detect a transient minor (several orders of magnitude lower than in LM differentiation) up-regulation of Hand1, Hopx, and Smad4 upon silencing of Fut1 (Fig. 4F) and suggest that "reduced α 1,2-fucosyl residues may facilitate BMP signals". It is an interesting hypothesis, but in my opinion the provided evidence is not sufficient to support it. I suggest the authors to perform a time-course analysis of Smad1/5/9 phosphorylation, e.g. by western blotting, upon Fut1 silencing (using siNT as control) to see if reduction of Fut1, and consequently α 1,2-fucosyl residues, lead to increase in phospho-Smad1/5/9 levels. Secondly, the authors could use LDN193189 (which they employed elsewhere in the manuscript) to test whether Hand1 and Hopx expression levels are not affected upon Fut1 silencing in the presence of the inhibitor.

Author response –

In response to this reviewer's kind suggestion, a time-course of Smad1/5/9 activity within silenced FUT1 and siNT cells after BMP4 stimulation revealed in western blotting an amplitude of Smad1/5/9 phosphorylation. In siNT cells, the first wave of oscillation for 24 hours was observed, whereas continuous SMAD activity was measured in FUT1-silenced cells over 24 hours. SMAD phosphorylation has oscillatory behavior; phosphorylation oscillations are tracked while cells are exposed to BMP4 (Miller et al., 2019). The minimum duration of BMP4 to trigger loss pluripotency-specific genes and upregulation of primitive streak markers is 30 min (Gunne-Bradden et al., 2020), and the levels of phosphorylated SMAD are affected by signal duration and ligand dosage (Miller et al., 2019). Furthermore, phosphorylated SMAD1/5 was discovered to target BMP-master genes in the nucleus of progenitor cells, facilitating lineage development, whereas targeting other BMP genes in the nucleus promoted cell proliferation in stem cells (Genander et al., 2015). This information may help to explain why BMP4 is constantly stimulated, followed by constitutive SMAD activation, and BMP-transcription factor increase in cells with low quantities of α -fucosyl groups, and why these functions are repressed in cells with high levels of α -fucosyl groups. The western blot results were added to **supplementary figure 6 C**.

We conducted the second experiment suggested by the reviewer, which involved testing HAND1 and HOPX in FUT1 silenced cells during 30 hours in media supplemented with the BMP signaling inhibitor, LDN193189. The results demonstrate that the HOPX gene expression level is similar in FUT1 silenced cells and control cells, and there are fewer HAND1 transcripts in siFUT1 cells than in siNT cells, showing that FUT1 silencing has no effect on HOPX, but HAND1 gene in siFUT1 cells may be under the control of LDN193189 that suppress its transcription, leading to lower mRNA production. These data were added to **supplementary figure 6 D**.

4. The authors claim that the reduction of α -fucosyl end groups after Bmp4 treatment (Fig. 6A) suggests that BMP4 is a "critical component in the regulation of sugar metabolism". In my opinion, the experimental evidence provided by the authors is not sufficient to prove a direct functional link between Bmp4 signaling and reduction of α -fucosyl end groups, but only shows that upon Bmp4-induced differentiation the levels of α -fucosyl end groups go down. Similar levels of α -fucose in hESCs and LDN193189 (Fig. 6A) is a result of LDN193189 blocking the differentiation, and do not prove direct link between Bmp4 activity and α -fucosyl end group levels.

α -fucosyl end groups, like Fut1, are reduced in all differentiation protocols tested by the authors, including neural differentiation, where Bmp4 is not active (Fig. 3J). Therefore, as the authors suggest earlier in the manuscript for Fut1, reduction of α -fucosyl groups appears to be a hallmark of early differentiation, not necessarily caused by Bmp4. Bmp4 induces exit from pluripotency (similarly to other cues the authors used other differentiation protocols), which in turn results in the down-regulation of Fut1 and α -fucosyl groups. Can Fut1 expression be controlled by pluripotency factors such as Oct4 or Nanog instead of Bmp4?

Author response –

The α -fucosyl end groups are reduced throughout all differentiation protocols, including neural differentiation, in which BMP4 is absent. We evaluated α -fucosyl end group reduction using each factor independently for 48 hours: bFGF, activin-A, CHIR, and BMP4. The most significant reduction of α -fucosyl end groups was observed in BMP4 then in bFGF and activin-A. In contrast, CHIR factor did not influence the level of α -fucosyl groups.

The last part in paragraph “Constitutive α 1,2-fucosyl GSL expression impairs hESC commitment” on page 12, is referred to the effects of factors other than BMP4 on FUT1 expression during differentiation – “In addition to LM markers, the expression of DE and NE genes and proteins (FOXA2, SOX17, PAX6) was slightly compromised in cells overexpressing FUT1 (Figure 5F, 5G, S5F, and S5G), but the levels of other tested genes (HHEX, SOX1, ZIC, OTX2) remained unchanged in the earlier days of differentiation (Figure S5H), suggesting that in addition to BMP, certain α 1,2-fucosyl structures can impact the signaling axis of receptors other than BMP, compromising a several of DE and NE downstream genes”.

We amended the text in paragraph “ α 1,2-fucosyl GSLs inhibit BMP signaling” on page 14, because these measures were not included in the results, and the reviewer's comment brought it to our attention.

Furthermore, the hESCs were treated with LDN193189, a BMP signaling inhibitor (not BMP4 inhibitor), to demonstrate that α -fucosyl end group expression remains unaltered when BMP signaling is suppressed. The reason we used LDN193189 is that it has an opposing impact than BMP4.

Regarding the reviewer's final question, OCT4 and other pluripotent markers likewise depend on BMP4 signaling (Gunne-Bradden et al., 2020).

5. The authors claim that α 1,2-fucosyl GSLs inhibit BMP4 signaling, based on the results of FACS analysis presented in Fig. 6E. The histogram displaying Fut1+ cells (Fig. 6E, middle panel) shows that even in Fut1+ cells, with high Fut1 and fucosyl group levels, a large fraction of Bmp4-treated cells is phospho-Smad1/5/8 positive. Also, this fraction appears larger than 10.57% as reported in the text. Could the quantification results shown in the

graphs in Fig. 6E perhaps be affected by the arbitrary setting of the gates?

The results presented in Fig. 6 only show that cells over-expressing Fut1 have a somewhat weaker response to Bmp4 signaling compared to the control cells. This may be caused by a number of factors, for example, lower levels of BMPRs in the FUT1⁺ cells or variations in membrane permeability between control and Fut1⁺ cells. In my opinion, the claim that α 1,2-fucosyl GSLs inhibit BMP4 signaling (title of the section) is a misinterpretation of the presented evidence.

To determine the effect of Fut1 over-expression on cellular response to Bmp4, I suggest the following experiment. The authors could quantitatively assess the levels of Smad1/5/8 phosphorylation by Western Blotting in a time-dependent manner (e.g., after 1, 2, 6, 12, 24 h of Bmp4 treatment) in control and Fut1⁺ cells. Analyzing these earlier time-points would also exclude any secondary effects that may have occurred within the 24 h period between Bmp4 application and cell collection in the presented experiment.

Author response –

In response to this reviewer's kind suggestion, a time-course of Smad1/5/9 activity within FUT1⁺ cells and control cells after BMP4 stimulation over 24 hours revealed in western blotting an amplitude of Smad1/5/9 phosphorylation like siNT cells. However, in this case, SMAD activity declined to low levels after 12 and 24 hours in control cells and terminated after 12 and 24 hours in FUT1⁺ cells (narrow amplitude).

According to Loh et al. (2014; 2016), BMP4 is essential for stem cell differentiation into mesoderm and is supplemented daily for the first three days of differentiation into lateral mesoderm. BMP signaling in lateral mesoderm cells increases different transcription factors based on the day of differentiation, and other relevant factors and inhibitors. The restricted amplitude of phosphorylated SMAD1/5/9 in Fut1⁺ cells affects downstream gene expression in the BMP signaling pathway. It inhibits the activation of SMAD proteins and their capacity to translocate to the nucleus and regulate gene transcription. This could affect the expression of target genes involved in cellular responses to BMP signaling. However, the specific consequences would depend on the context and the specific regulatory mechanisms involved. FUT1 and α -fucosyl end groups inhibit BMP signaling but do not prohibit it, compromising transcription factor expression in the lateral mesoderm. These data were added to **supplementary figure 6 B**.

We also corrected the data given in the **Fig 6E graph**; FACS analysis was impacted by the arbitrary gate setup.

We'd like to persuade the reviewer on the relationship between FUT1 and BMP4 or BMP signaling. Models of the formation of human embryos have recently been developed through the culture of stem cells, which show the extraembryonic layers, the beginning of embryo formation within these layers, and provide reliable information about the early post-implantation embryonic compartments and their relationships (Weatherbee et al. 2023; Pedroza et al. 2023; Oldak et al. 2023). One of the signaling pathways found essential to achieve an organized structure of the human embryo-like models is the BMP signaling. Amnion, primordial germ cells and extraembryonic mesenchyme in the developed human embryo-like models differentiate in response to BMP4. Based on these models, we analyzed published single-cell RNA-sequencing raw data for human embryo-like models (series accession number GSE208195 in the Gene Expression Omnibus) (Pedroza et al. 2023) and discovered an expression of BMP4 gene in amnion and extraembryonic mesenchyme clusters (**Figure 1A-B in this letter**, adopted from (Pedroza et al. 2023)) and a negligible expression of FUT1 genes (**Figure 1C in this letter**, created by BGU bioinformatics unit). In 2D cultures, FUT1⁺ cells and hESCs were differentiated into amnion cells for 4 days; Representative brightfield images demonstrating circle-like amnion cell morphology produced by WT cells, and unshaped structure formed by the FUT1⁺ cells (**Figure 1D in this letter**, performed by Michal Nagar from my lab, scale bar 400 μ m). These preliminary results from extraembryonic lineages may show that cells producing and depending on BMP4 lack

FUT1, and cells expressing FUT1 have difficulty forming circle-like amnion cell shape in media supplemented with BMP4.

Figure for referees not shown.

Minor comments:

1. In Fig. 4F expression of genes is assayed at different time points. For example, Hand1 at 54 and 78h. Isl1 and Nkx2.5 at 30h. Isl1 and especially Nkx2.5 are expected to be up-regulated at the same/similar time point as Hand1. Were the expression of Nkx2.5 and Isl1 assayed at 54 and 78h as well? If so, could the authors include the results in the figure? Otherwise, could the authors explain why different time point were chosen.

Author response –

The reviewer is right, the Nkx2.5 TF is expected to be up regulated at the same/similar time point (54h and 78 h) as Hand1 TF. NKx2.5 gene did not up-regulate, we mentioned it in the text – “In contrast, downregulation of FUT1 in hESCs led to a transient upregulation of lateral mesoderm (LM; cardiac) specific genes, HAND1, HOPX, and SMAD4, except the NKX2.5 gene, in both the mRNA and protein levels for 30-102 h”. The acquired results had a substantial inaccuracy and were consequently excluded from the results.

2. In the Discussion section the authors talk about the activation of Smad2/3 when α 1,2-fucosyl glycans are over-expressed. This interesting hypothesis can be relatively easily checked using Western blotting, but in the experimental section I did not see any analysis of

Smad2/3 phosphorylation. I suggest the authors either perform this experiment or exclude the speculation of Smad2/3 activation from the Discussion.

Author response –

The discussion was revised according to both reviewer's suggestions. Your comments improve the manuscript.

3. Fig. 4E. Only Nanog expression is shown at 30h and 54h. Oct4 and Sox2 expressions are only shown at 30h. What does Oct4 and Sox2 expression at 54h look like?

Author response –

The TFs, OCT4 and SOX2 have recovered following the initial change we measured. This was written in the discussion and point 2 in the major comment

4. Fig. 4F. For all genes except for Smad4 the p values are calculated for siNT vs siFUT1. Is the difference between Smad4 expression in siNT and siFUT1 statistically significant? Similarly, Fig. S4E should show p values for siNT vs siFUT1.

Author response –

We add p values to Figs 4F and S4E.

5. There are some inconsistencies in terminology throughout the manuscript. For example: α 1,2-fucosyl vs α 1,2-fucosyl (extra space); globoseries vs globo-series; Oct3\4 (Fig. 4E) vs Oct3/4 (Fig. 3I), etc.

Author response –

The inconsistencies in terminology were corrected.

Dear Dr. Lichtenstein,

Thank you for the submission of your revised manuscript to our editorial offices. I have now received the reports from the two referees that I asked to re-evaluate the study, you will find below. As you will see, both referees now support the publication of the study in EMBO reports. Referee #2 has some remaining concerns and suggestions to improve the manuscript, I ask you to address in a final revised manuscript. Please also provide a final p-b-p-response regarding the remaining points of the referee.

- Please provide a final title with not more than 100 characters (including spaces) and without colon.
- Please reduce the number of words in the abstract to 175 and provide it written in present tense.
- Please provide individual production quality figure files as .eps, .tif, .jpg (one file per figure), of main figures and EV figures (see below). Please upload these as separate, individual files upon re-submission. Please note that each figure should have only one page!

The Expanded View format, which will be displayed in the main HTML of the paper in a collapsible format, has replaced the Supplementary information. You can submit up to 6 images as Expanded View. Please follow the nomenclature Figure EV1, Figure EV2 etc. The figure legend for these should be included in the main manuscript document file in a section called Expanded View Figure Legends after the main Figure Legends section. Additional Supplementary material should be supplied as a single pdf file labeled Appendix. The Appendix should have page numbers and needs to include a table of content on the first page (with page numbers) and legends for all content. Please follow the nomenclature Appendix Figure Sx, Appendix Table Sx etc. throughout the text, and also label the figures and tables according to this nomenclature.

- We now use CRediT to specify the contributions of each author in the journal submission system. CRediT replaces the author contribution section. Please use the free text box to provide more detailed descriptions and do NOT provide your final manuscript text file with an author contributions section. See also our guide to authors:
<https://www.embopress.org/page/journal/14693178/authorguide#authorshipguidelines>

- We updated our journal's competing interests policy in January 2022 and request authors to consider both actual and perceived competing interests. Please review the policy <https://www.embopress.org/competing-interests> and update your competing interests if necessary. Please name this section 'Disclosure and Competing Interests Statement' and put it after the Acknowledgements section.

- Please order the manuscript sections like this, using these names:

Abstract - Keywords - Introduction - Results - Discussion - Methods - Data availability section - Acknowledgements - Disclosure and Competing Interests Statement - References - Figure legends - Expanded View Figure legends

- The "Data Availability section" should be dedicated to datasets produced in this study deposited at external repositories. Please restrict this section to such datasets and do not format it as a table. See also:

<http://embor.embopress.org/authorguide#datadeposition>

I suggest using the model below:

Data availability Section

- All previously published datasets that were re-used in the study should be cited and called out as have *data citations* in the reference list. Data citations in the article text are distinct from normal bibliographical citations and should directly link to the database records from which the data can be accessed. In the main text, data citations are formatted as follows: "Data ref: Smith et al, 2001" or "Data ref: NCBI Sequence Read Archive PRJNA342805, 2017". In the Reference list, data citations must be labeled with "[DATASET]". A data reference must provide the database name, accession number/identifiers and a resolvable link to the landing page from which the data can be accessed at the end of the reference. Further instructions are available at: <http://www.embopress.org/page/journal/14693178/authorguide#referencesformat>

- Please make sure that the number "n" for how many independent experiments were performed, their nature (biological versus technical replicates), the bars and error bars (e.g. SEM, SD) and the test used to calculate p-values is indicated in the respective figure legends (also for potential EV figures and all those in the final Appendix). Please also check that all the p-values are explained in the legend, and that these fit to those shown in the figure. Please provide statistical testing where applicable. Please avoid the phrase 'independent experiment', but clearly state if these were biological or technical replicates. Please also indicate (e.g. with n.s.) if testing was performed, but the differences are not significant. In case n=2, please show the data as separate datapoints without error bars and statistics. See also: <http://www.embopress.org/page/journal/14693178/authorguide#statisticalanalysis>

If n<5, please show single datapoints for diagrams. Presently, some diagrams seem to miss the 'n.s.'. Please check. Moreover:

- Please note that the supplementary figure panels 6a-d are not labelled in the figure, however the corresponding legends are labelled as 6a-d. This needs to be rectified.
- Further, the legends for these figures don't seem to match the figures provided. Please check.
- Please indicate the statistical test used for data analysis in the legend of supplementary figure 2.
- Please note that in figures 6a-e; there is a mismatch between the annotated p values in the figure legend and the annotated p values in the figure file that should be corrected.
- Please note that information related to n is missing in the legends of figures 3f, h; 5g-h; 6a-b, supplementary figures 3a-b.
- Although 'n' is provided, please describe the nature of entity for 'n' in the legends of figures 3i-j, m.
- Please note that the error bars are not defined in the legends of figures 6a-b.

- Please add to each legend (main, and EV figures, where applicable) a 'Data Information' section explaining the statistics used or providing information regarding replicates and scales. See:

- Please remove the tables from the methods section and show this information only in the reagents and tools table. Please make sure the reagents and tools table is called out where needed.

- Please use our reference format:

- Per journal policy, we do not allow 'data not shown', which is stated several times in the manuscript. All data referred to in the paper should be displayed in the main or Expanded View figures, or an Appendix. Thus, please add these data (or change the text accordingly if these data are not central to the study). See:

<https://www.embopress.org/page/journal/14693178/authorguide#unpublisheddata>

- Please make sure that all the funding information is also entered into the online submission system and that it is complete and similar to the one in the acknowledgement section of the manuscript text file.

- Please make sure that all figure panels are called out separately and sequentially. Presently, there seems to be no callout for panel 5l. Please check.

- Thanks you for depositing the source data (SD) for this study. But please provide a filled in source data checklist with your final submission and make sure alle SD requested is provided.

In addition, I would need from you:

Best,

Referee #1:

The manuscript is suitable for publication in EMBO reports without further revision.

Referee #2:

I would like to thank the authors for addressing the raised comments. The manuscript has improved. There are a few remaining points to consider.

1. In their response, the authors agree that there is no evidence supporting the role of Fut1 activity or alpha1,2-fucosyl GSLs in pluripotency. Likewise, they mention their "in vitro observations that Fut1 is not a hallmark of pluripotency".

Accordingly, in the Discussion the authors have changed the sentence "Our observations provide evidence for the high level of alpha1,2-fucosyl glycoconjugates required to maintain the pluripotency state and for the low level needed to allow cell differentiation." to "Our observations support the high amount of alpha1,2-fucosyl glycoconjugates in pluripotency state and the low level required for proper cell differentiation."

However, the results section still contains statements such as "FUT1 gene upregulation and its enhanced activity are one of the hallmarks of pluripotency", "FUT1 but not its paralog is presumably a crucial enzyme required in the pluripotency state" and a few others. Is this an oversight and if so, should these statements be rephrased or removed?

2. Inhibition of BMP signaling by alpha1,2-fucosyl GSLs is one of the major findings reported in the manuscript.

I would like to thank the authors for performing the time-course western blotting experiment to assess the effect of Fut1 on Bmp4 signaling (Fig. S6B), but have a few concerns about its interpretation.

The authors suggest that FUT1+ ECs display "lower quantity of phosphorylation than that found in control ECs". However, due to high background signal and uneven loading (based on the Actin signal intensities), the difference in phospho-Smad1/5/8 levels between FUT1+ and control ECs is not obvious.

Did the authors quantify the intensities of the pSmad1/5/8 bands relative to the loading control? And was the experiment performed using biological replicates? Quantification of bands from biological replicates would help making a statistically significant conclusion.

3. Related to the previous point. Fig. 6E shows that 24 hours of Bmp4 treatment results in phospho-Smad1/5/8 detection in over 30% of control ECs and in about 6% of FUT1+ECs (5 fold difference). Yet, Fig. S6B shows almost no pSmad1/5/8 after 24 hours of Bmp4 treatment in both control ECs and FUT1+ECs. I was wondering if the experimental conditions (cells and 24 hours of Bmp4) were the same for the experiments shown in Fig. 6E and Fig. S6B. If they were the same, could the authors discuss the apparent discrepancy? If the conditions were different, could it be indicated in the text, methods, or legends?

4. In the sentence "A wave of phosphorylated SMAD 1/5/8 oscillation was observed in siNT cells for 24 h, whereas siFUT1 cells continuously phosphorylated the SMAD 1/5/8 protein for 24 h (Figure S6B).", should it be Figure S6C instead of Figure S6B?

5. The color bar for the heatmap presented in Figure 6D seem to have wrong colors for the down-regulated genes.

EMBOR-2023-58438-T

Dear editor and reviewers,

A complete, point-by-point response to the second letter is included below. Sentences and paragraphs added to correct faults in the amended version are underlined yellow.

Detailed point-by-point response to reviewer's comments**Referee #1:**

The manuscript is suitable for publication in EMBO reports without further revision.

Author response:

We'd like to thank the reviewer.

Referee #2:

I would like to thank the authors for addressing the raised comments. The manuscript has improved. There are a few remaining points to consider.

Author response:

We'd like to thank the reviewer for her/his suggestions for improving our manuscript.

1. In their response, the authors agree that there is no evidence supporting the role of Fut1 activity or alpha1,2-fucosyl GSLs in pluripotency. Likewise, they mention their "in vitro observations that Fut1 is not a hallmark of pluripotency".

Accordingly, in the Discussion the authors have changed the sentence "Our observations provide evidence for the high level of α 1,2-fucosyl glycoconjugates required to maintain the pluripotency state and for the low level needed to allow cell differentiation." to "Our observations support the high amount of α 1,2-fucosyl glycoconjugates in pluripotency state and the low level required for proper cell differentiation."

However, the results section still contains statements such as "FUT1 gene upregulation and its enhanced activity are one of the hallmarks of pluripotency", "FT1 but not its paralog is presumably a crucial enzyme required in the pluripotency state" and a few others. Is this an oversight and if so, should these statements be rephrased or removed?

Author response:

In response to the reviewer's feedback, we went through the result section and edited any sentences that did not reflect the findings we gave.

2. Inhibition of BMP signalling by alpha1,2-fucosyl GSLs is one of the major findings reported in the manuscript.

I would like to thank the authors for performing the time-course western blotting experiment to assess the effect of Fut1 on Bmp4 signaling (Fig. S6B) but have a few concerns about its interpretation.

The authors suggest that FUT1+ ECs display "lower quantity of phosphorylation than that found in control ECs". However, due to high background signal and uneven loading (based on the Actin signal intensities), the difference in phospho-Smad1/5/8 levels between FUT1+ and control ECs is not obvious.

Did the authors quantify the intensities of the pSmad1/5/8 bands relative to the loading control? And was the experiment performed using biological replicates? Quantification of bands from biological replicates would help making a statistically significant conclusion.

Author response:

Following the reviewer's critique, we analyzed the band intensities by using Image J software and revised the sentence to – "When FUT1⁺ECs were activated with BMP4, a similar single oscillation was seen; the high phosphorylation level was reached after 1 h and eventually dropped to lower levels than phosphorylation in control ECs" The results are present in Fig. EV5B.

The experiment was performed twice (biological replicates). The background is the consequence of washing the blots in a base solution after actin staining, which was followed by staining for Smad1/5/8 phosphorylation. The band intensities were estimated by normalizing actin bands to actin bands at time zero, normalizing pSMAD bands to pSMAD bands at time zero, and then dividing the normalized pSMAD values by normalized actin values. The M&M now includes a description of how band intensity is calculated.

3. Related to the previous point. Fig. 6E shows that 24 hours of Bmp4 treatment results in phospho-Smad1/5/8 detection in over 30% of control ECs and in about 6% of FUT1+ECs (5-fold difference). Yet, Fig. S6B shows almost no pSmad1/5/8 after 24 hours of Bmp4 treatment in both control ECs and FUT1+ECs. I was wondering if the experimental conditions (cells and 24 hours of Bmp4) were the same for the experiments shown in Fig. 6E and Fig. S6B. If they were the same, could the authors discuss the apparent discrepancy? If the conditions were different, could it be indicated in the text, methods, or legends?

Author response:

The flow Cytometry analyses individual cells, providing a distribution of phosphorylation levels across the cell population. This can reveal subpopulations with high phosphorylation that would be averaged out in a Western blot. The western blot measures the average phosphorylation level across the entire cell population, which might dilute the signal if only a subset of cells is highly phosphorylated. Fig 6E presents a subpopulation with high SMAD1/5/8 phosphorylation and Fig EV5B (was Fig S6B) presents average SMAD1/5/8 phosphorylation in the total cell proteins.

After quantifying the band intensities, the observed differences 24 hours after Bmp4 treatment result in a 4-fold increase in phospho-Smad1/5/8 in control ECs compared to FUT1+ ECs.

4. In the sentence "A wave of phosphorylated SMAD 1/5/8 oscillation was observed in siNT cells for 24 h, whereas siFUT1 cells continuously phosphorylated the SMAD 1/5/8 protein for 24 h (Figure S6B).", should it be Figure S6C instead of Figure S6B?

Author response:

Yes. The reviewer is right. Thank you.

5. The color bar for the heatmap presented in Figure 6D seem to have wrong colors for the down-regulated genes.

Author response:

The heatmap was performed by OriginLab software, <https://www.originlab.com/>

Dr. Rachel Lichtenstein
Ben Gurion University
Biotechnology Engineering
P.O. Box 653
Beer Sheva, Israel 84105
Israel

Dear Dr. Lichtenstein,

I am very pleased to accept your manuscript for publication in the next available issue of EMBO reports. Thank you for your contribution to our journal.

Yours sincerely,
